# Lipid dependence of connexin-32 gap junction channel conformations

Pia Lavriha[1,2], Carina Fluri[2], Jorge Enrique Hernández González[3] ✉ & Volodymyr M. Korkhov [1,2] ✉

Connexin-32 (Cx32) gap junction channels (GJCs) mediate intercellular coupling in various tissues, including myelinating Schwann cells. Mutations in Cx32, such as W3S, are associated with X-linked Charcot-Marie-Tooth (CMT1X) disease. Lipids regulate Cx32 GJC permeation, although the regulatory mechanism is unclear. Here, we determine the cryo-EM structures of Cx32 GJCs reconstituted in nanodiscs, revealing that phospholipids block the Cx32 GJC pore by binding to the site formed by N-terminal gating helices. The phospholipid-bound state is contingent on the presence of a sterol molecule in a hydrophobic pocket formed by the N-terminus: the N-terminal helix of Cx32 fails to sustain a phospholipid binding site in the absence of cholesterol hemisuccinate. The CMT1X-linked W3S mutant which has an impaired sterol binding site adopts a conformation of the N-terminus incompatible with phospholipid binding. Our results indicate that different lipid species control connexin channel gating directly by influencing the conformation of the N-terminal gating helix.

Cell communication is crucial for coordination of cells with their environment and generation of appropriate cellular responses. Among different means of cellular communication, gap junction intercellular communication (GJIC) enables a fast and direct signal exchange between neighboring cells and allows electrical and metabolic coupling of the tissues[1,2]. In vertebrates GJIC is mediated by connexins, which assemble into gap junction channels (GJCs), that contain a central pore, connecting the cytoplasms of two neighbor cells and allowing solute permeation from one cell into the other[1–3]. Each of the 21 connexin isoforms in humans has a distinct expression pattern, permeability, and gating properties, establishing the required GJIC between specific cell types[4]. The GJIC has to be tightly regulated to ensure the passage of appropriate solutes and to prevent the exchange of noxious signals between coupled cells[2–4]. Among the regulators, lipids (fatty acids, sterols, and phospholipids) have gained increased attention for their effects on GJC permeability. These hydrophobic small molecules have been shown to alter GJC permeability either directly by binding to the channel, or indirectly

via changes in the biophysical properties of the membrane or via activation of distinct signaling pathways[5–8].

The potential role of direct lipid-protein interactions with the connexin channels has been illustrated by the recent structural studies of connexin hemichannels (HCs) and GJCs. Ordered lipid-like molecules have been identified at five different locations in recent cryo-electron microscopy (cryo-EM) structures of connexin HCs and GJCs: (i) as annular lipids, lining the extracellular lipid leaflet side of the channel (Cx46/50 GJC[9], Cx43 GJC[10,11], Cx36 GJC[12], and Cx32 HC and GJC[13]), (ii) below the N-terminal helix (NTH) (Cx43 GJC[10,11], Cx43 HC[11], Cx32 HC[13]), (iii) between NTHs (Cx43 GJC[10,11]), (iv) lining the cytoplasmic entry of the pore (Cx36 GJC[12]), and (v) lining the inside of channel pore (Cx31.3 HC[14], Cx36[12], Cx46/50[9], Cx32 GJC and HC[13] and Cx43 GJCs[10,11]). The presence of these ordered lipid-like small molecules around or inside the channels suggests that lipids may have a direct effect on connexin channel structure and function.

Connexin-32 (Cx32) is a widely expressed isoform of the connexin channel family, found abundantly in the liver cells from which

[1]Laboratory of Biomolecular Research, Paul Scherrer Institute, Villigen, Switzerland. [2]Institute of Molecular Biology and Biophysics, ETH Zurich, Switzerland. [3]Department of Physics, Institute for Biosciences, Letters and Exact Sciences, Sao Paulo State University, São José do Rio Preto, Brazil. ✉e-mail: jorge.hernandez@unesp.br; volodymyr.korkhov@psi.ch

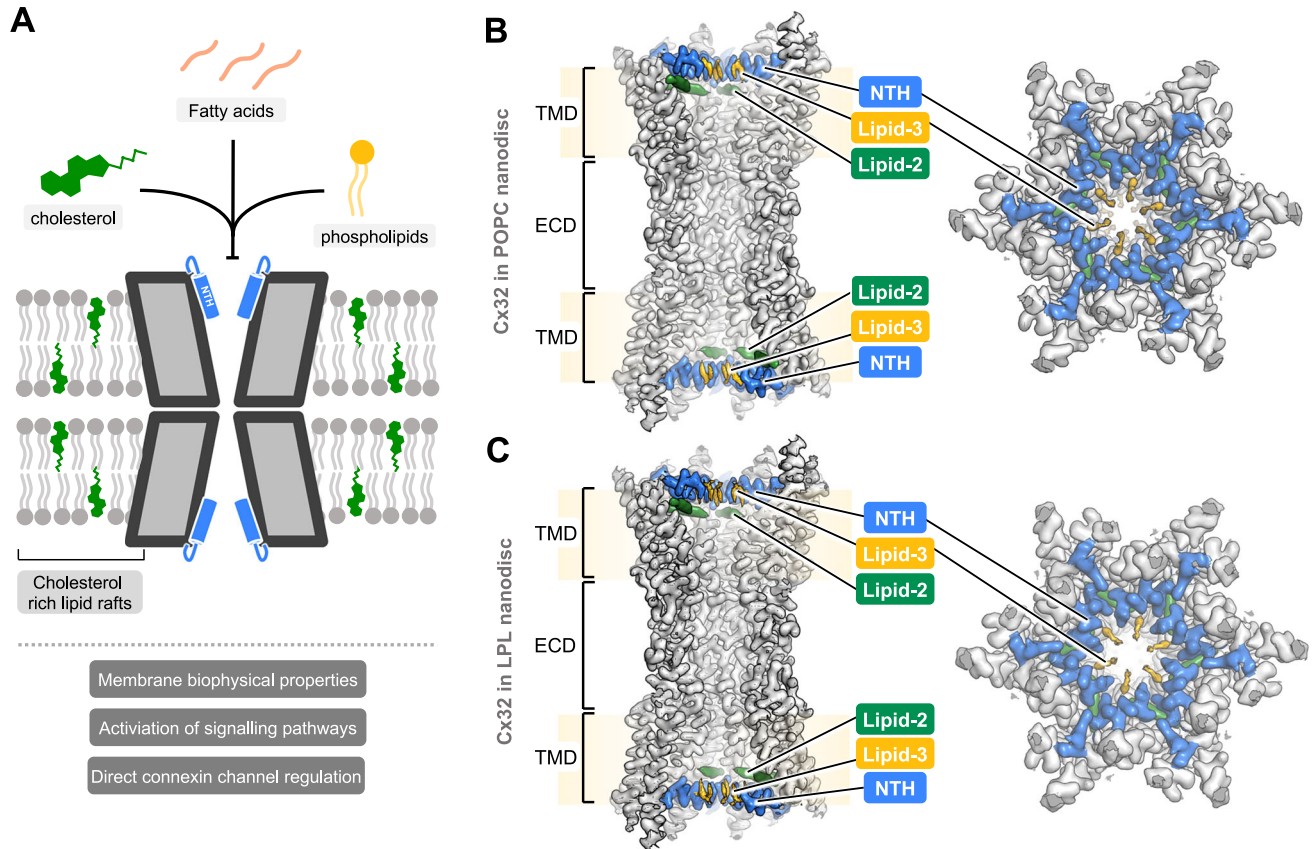

**Fig. 1 | Cryo-EM structures of Cx32 GJC in nanodisc. A** Schematic of the functional effects of lipids on Cx32 GJC. Structures of Cx32 GJC, reconstituted in nanodiscs, containing POPC (**B**) or LPL (**C**), show an ordered N-terminal helix (NTH), and the presence of two densities, likely corresponding to lipids, lipid-2 and lipid-3. TMD transmembrane domain, ECD extracellular domain.

it was originally isolated[15,16]. Cx32 is also found in the myelinating Schwann cells, where it appears to be important for maintaining the cellular homeostasis, with roles for both HCs and GJCs[17]. Multiple mutations in Cx32 are associated with the X-linked Charcot-Marie-Tooth disease (CMT1X), a disorder accompanied by demyelination of the peripheral neurons and eventually leading to degeneration of the muscles, for which there is currently no cure[17–20]. Cx32 HCs play a role in secreting ATP at the Schwann cell surface[21], and the GJCs are important to maintain lateral connections across the myelin layer[22]. We have recently determined the structures of human Cx32 HCs and GJCs in detergent micelles, where we observed a conformation of the N-terminal gating helix of Cx32 supported by a lipid-like molecule in the lipid-1 site, which we interpreted as a molecule of cholesterol hemisuccinate (CHS)[13]. Our work showed that two CMT1X-linked mutants, W3S and R22G, have a structural and a functional defect in the HCs, with little or no observable structural effects on Cx32 GJCs, and no measurable effects on Cx32-mediated GJC activity in cell-based assays[13].

In addition to our direct observation of sterol-like molecules bound to Cx32, the role of lipids in Cx32 regulation has been explored previously (Fig. 1A). Cx32 preferentially resides in cholesterol-rich caveolin-containing lipid rafts[8]. Phospholipids also directly associate with Cx32 and together with the sterols influence Cx32 HC permeability[6]. However, how phospholipids and sterols interact with Cx32 channels in the context of a lipid bilayer remains unclear.

In this work, to investigate the effect of phospholipids and sterols on Cx32 channels, we reconstituted the purified Cx32 channels into lipidic environment in nanodiscs, and determined their cryo-EM structures.

## Results

### Structures of Cx32 GJCs in lipid environment

To address the effect of different lipids on Cx32 GJC, we purified Cx32 in digitonin, and reconstituted the protein into 1-palmitoyl-2-oleoyl-sn-*glycero*-3-phosphocholine (POPC) and liver polar lipids (LPL) -containing nanodiscs (Fig. S1A, B). We chose two different lipid preparations for the following reasons: in the case of POPC we would obtain nanodiscs with clearly defined lipid content, and LPL would represent a close to physiological lipid composition, since Cx32 is known to be abundantly expressed in hepatocytes[23]. We prepared samples for cryo-EM by plunge-freezing the purified and reconstituted samples in liquid ethane, collected the cryo-EM data, and determined the structures of Cx32 GJC in nanodiscs, at 3.20 Å and 3.29 Å resolution for POPC and LPL, respectively (Figs. S2, S3 and S4A, B, Table 1). The structures were obtained by imposing the D6 symmetry. Refining the particles without imposing symmetry (C1) resulted in a similar reconstruction, albeit at a slightly lower resolution (Figs. S2–3, S5–6).

Overall, the structures of Cx32 GJC in POPC- and LPL-containing nanodiscs are very similar to each other, with an RMSD of 0.495 Å (Fig. 1B, C, Fig. S6), and each was overall similar to the structure of Cx32 GJC in detergent[13]. The four transmembrane helices (TM1-4) and the extracellular loops 1 and 2 (ECL1-2) of individual connexin subunits have the same conformation in the presence or absence of lipids, indicating that lipids do not affect the assembly of two HCs into a GJC, and do not affect the overall pore architecture. These similarities apply also to the orientation of the individual amino acid residues, with an RMSD of 0.259 Å and 0.636 Å between Cx32 GJC in detergent, and Cx32 GJC in POPC- and LPL-containing nanodiscs, respectively.

Nonetheless, the addition of lipids has a notable effect on the conformation of the Cx32 N-terminal helix (NTH), an α-helical region

**Table 1 | Cryo-EM data collection, data processing, and model building parameters**

| Data collection | | | | |
|---|---|---|---|---|
| Sample | Cx32 GJC POPC | Cx32 GJC LPL | W3S GJC POPC | Cx32 GJC no CHS POPC |
| Instrument | FEI Titan Krios/Gatan K3 Summit/Quantum GIF | | | |
| Voltage [kV] | 300 | | | |
| Electron dose [e-/Å] | 61.6 | Dataset 1: 61.2 | Dataset 2: 50 | 55        55 |
| Defocus range [μm] | −0.5 to −2.5 | | | |
| Pixel size [Å] | 0.6506 | 0.6506 | 0.6609 | 0.651 |
| Map resolution [Å] FSC 0.143 | 3.16 | 3.29 | 2.35 | 3.12 |
| Number of particles | 80,708 | 53,450 | 68,649 | 15,537 |
| **Refinement** | | | | |
| Model resolution [Å] FSC 0.5 | 3.1 | 3.3 | 2.3 | 3.1 |
| Map sharpening B-factor [Å] | −50 | −50 | −20 | −20 |
| Map CC | 0.92 | 0.92 | 0.91 | 0.89 |
| **Model** | | | | |
| Protein residues/ ligand | 2352/24 | 2352/24 | 2316/0 | 2256/0 |
| ADP (B-factor), protein | 34.73 | 54.73 | 68.25 | 85.96 |
| ADP (B-factor), ligand | 89.58 | 107.79 | – | – |
| Bond length r.m.s.d. (Å) | 0.002 | 0.003 | 0.003 | 0.003 |
| Bond angles r.m.s.d. (°) | 0.417 | 0.477 | 0.480 | 0.474 |
| **Validation** | | | | |
| MolProbity score | 1.12 | 1.52 | 1.02 | 1.50 |
| Clash score | 2.86 | 4.34 | 2.39 | 3.51 |
| Rotamer outliers (%) | 1.14 | 2.27 | 0 | 2.96 |
| **Ramachandran plot** | | | | |
| Favored (%) | 98.96 | 97.83 | 98.63 | 98.37 |
| Allowed (%) | 1.04 | 2.17 | 1.37 | 1.63 |
| Disallowed (%) | 0 | 0 | 0 | 0 |

critical for Cx32 gating[13,24,25] (Figs. 2–4). The NTH transitions from a flexible structural element unresolved in our cryo-EM reconstruction of Cx32 GJC in detergent[13] to a well-ordered α-helical conformation, as well as moves from lining the channel pore to pointing towards the symmetry axis of the pore. This movement is independent from conformational changes in the TM1-4 or ECL1-2. This conformation resembles the NTH conformation of Cx32 HC that we previously determined using Cx32 samples in detergent[13]. Performing protomer focused classification (PFC) on the individual subunits of Cx32 GJC in nanodiscs did not expose any substantial variability in the NTH conformation (Fig. S7A, B), indicating that the NTH of all subunits in these conditions assume a similar conformation.

It is noteworthy that, unlike the Cx32 sample that previously led to cryo-EM reconstructions of both GJCs and HCs in detergent micelles[13], upon nanodisc reconstitution we could observe predominantly the full GJCs. This may reflect a greater stability of the fully assembled GJCs, allowing these complexes to stay intact during detergent removal. In contrast, HCs may suffer in a number of ways during the lipid reconstitution procedure, which results in very few particles that can be used

for cryo-EM image processing and structure determination of HCs in nanodiscs. Therefore our study was focused on the Cx32 GJCs.

**Annular and pore-lining lipids**

Several lipid-like densities have been identified in the Cx32 GJC structure in detergent. In particular, lipid-1, which lines the inside of the channel pore, and annular lipids on the extracellular membrane leaflet side of the channel. Due to the limited resolution of the 3D reconstructions, the lipid identities could not be unequivocally determined[13]. The densities at equivalent positions in Cx32 GJC structure in nanodisc are resolved better and could correspond to acyl chains (Fig. 2A). Acyl chains in similar positions were identified in the structures of Cx46/50 GJC[9], Cx43 GJC[10], and Cx36 GJC[12]. As we were not able to determine whether these acyl chains represent free fatty acids copurified with the protein, or whether they are parts of bound phospholipids, we left these regions of the EM maps unmodelled (the hexadecane in Fig. 2 is shown for illustration purposes).

Annular lipids are present at the interface between two neighboring connexin subunits (Fig. 2B). The lipid binding pocket is formed by TM1 and TM4 of one subunit and TM2 and TM3 of the neighboring subunit. The pocket is formed predominantly by hydrophobic amino acids. Similarly, lipid-1 binds to a hydrophobic pocket formed by neighboring connexin subunits, in this case formed by TM1 of one and TM1 and TM2 of the second (Fig. 2C). The lipid-1 density spans the majority of the transmembrane domain, from lipid-2 below the NTH to the interface between the transmembrane and extracellular domains.

**Sterol-like and phospholipid-like molecules stabilize the NTH**

Closer inspection of the density map around the NTH in both Cx32 GJC nanodisc structures, reveals two densities which could stabilize the observed NTH conformation (Fig. 2A). The first is positioned below the NTH (Fig. 2D) and is shaped like a sterol. This density is similar to the one we have observed at the lipid-2 site previously for Cx32 HCs in detergent[13], as well as in the structures of Cx43 GJCs solved in both nanodiscs and in detergent[10,11]. However, the identity of the lipid-2 molecule could not be determined definitively. Although we modeled a cholesterol molecule into the corresponding part of the density map, this density could also accommodate other sterols, such as cholesteryl hemisuccinate (CHS) and digitonin, which were used during stages of protein purification, as described in "Methods" (Fig. S8A). This binding site is hydrophobic, with the W3 residue side chain in direct proximity, participating in the interactions with the bound lipid molecule. Residue W3 is conserved in all β-group connexins (Fig. 2F), suggesting its functional importance in interactions with the lipid-2 as a mediator of NTH conformational changes in Cx32 and other GJCs.

An additional observed density, which we named lipid-3, is located between the neighboring NTH regions. The lipid-3 density has a similar shape in both POPC- and LPL-containing Cx32 GJC nanodisc structures (Fig. 2E, Fig. S8B, C). The density is consistent with the glycerophosphoric segment and for parts of the two aliphatic chains of POPC. The binding site for lipid-3 is formed by residues M1, N2, G5 and L9 of one Cx32 subunit and M1 of the neighboring subunit. The position equivalent to G5 in other β-group connexin isoforms is not conserved but occupied by hydrophobic residues (Fig. 2F), which could mediate the lipid-3 interaction in other β group connexins. Mutation of one of the conserved residues, L9W, is associated with CMT1X[26], indicating the functional importance of this site. While our interpretation of the observed densities is that they correspond to the bound phospholipid molecules, we can not completely exclude a possibility that these densities may also correposnd to bent acyl chain of the lipids, or even some other small molecule species present in the sample (co-eluting with the purified protein and appearing as ordered density upon addition of the lipids and removal of the detergent).

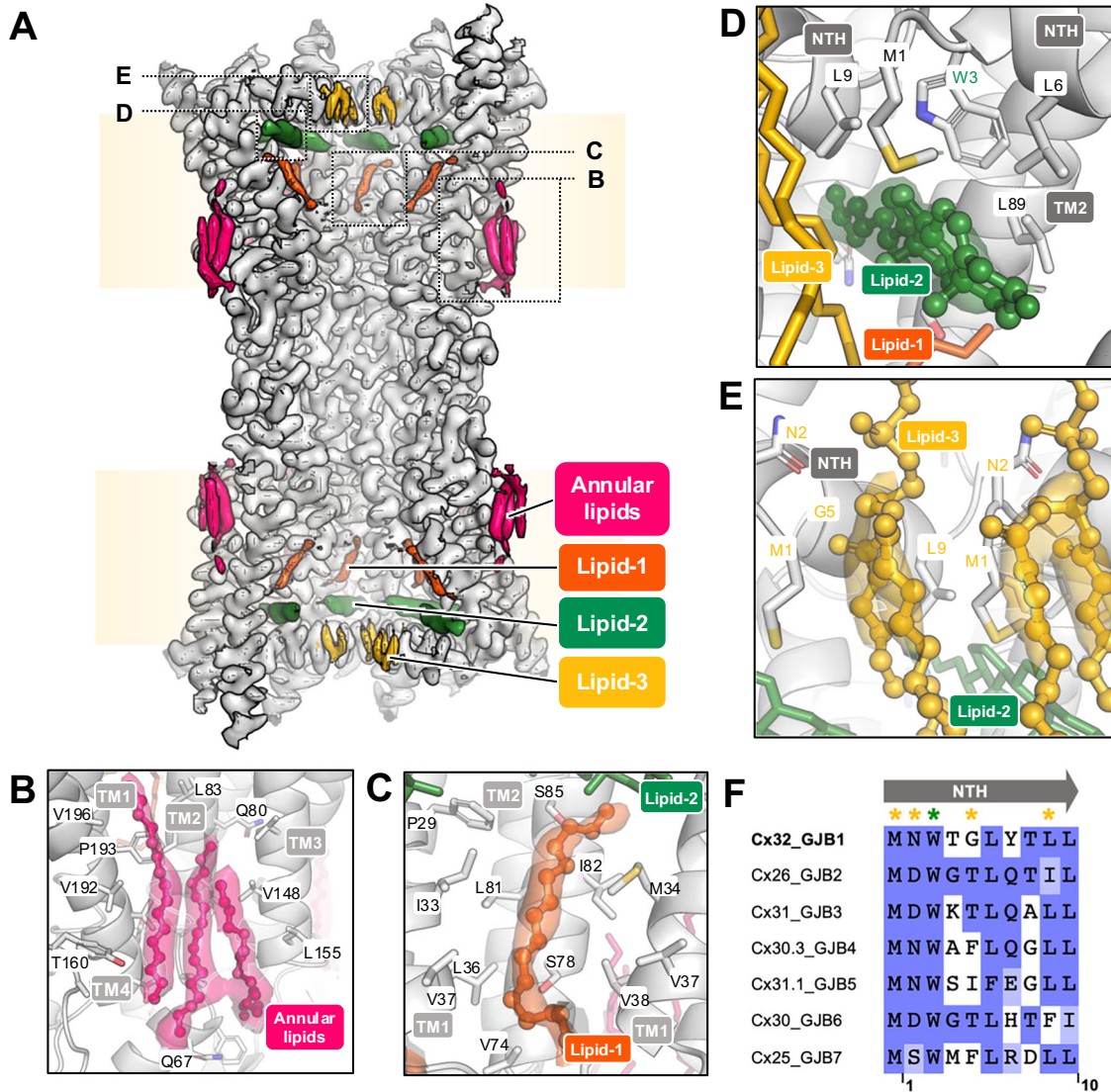

**Fig. 2 | Lipids bind to Cx32 GJC in nanodisc. A** Lipid densities in Cx32 GJC nanodisc structure (POPC-containing nanodiscs). Pale yellow boxes correspond to the approximate lipid bilayer boundaries. Dashed boxes represent the positions of lipid densities, represented in panels (**B**–**E**). The binding sites of annular lipids (**B**), lipid-1 (**C**), lipid-2 (**D**) and lipid-3 (**E**). **F** Multiple sequence alignment of β-group connexins, performed using Clustal Omega algorithm. The alignment is colored by BLOSUM62 color scheme. Yellow asterisks represent the putative lipid-3-interacting residues, and green asterisk indicates the W3 residue, interacting with lipid-2.

## Cx32 reconstituted in nanodiscs is effectively plugged by lipid molecules

The structure of Cx32 GJC reconstituted into nanodiscs shows a conformational change of the NTH that leads to a decrease in the pore diameter from approximately 15 Å in an open conformation to 12 Å (Fig. 3A, B, Fig. S8D). This NTH reorganization modifies the electrostatic potential of the cytoplasmic pore entry, which becomes more positively charged (Fig. 3C). We refer to this Cx32 GJC conformation as the ordered NTH state (NO-state). Including lipid-3 in pore diameter calculations shows an additional restriction of the pore diameter to only 4 Å (Fig. 3A, B), which effectively constricts and blocks the pore entry to ions or small molecules.

## Lipid-2 binding is required for NTH stabilization

The positions of lipid-2 and lipid-3 in the vicinity of the Cx32 NTH regions, known to be the key gating regions of connexin channels, suggest that these two lipids may stabilize NTH in the NO-state. To assess the requirement for lipid-2 and lipid-3 in NTH stabilization, we determined the GJC structures of (i) Cx32 GJC purified without addition of the cholesterol analog CHS during protein purification, and (ii)

Cx32 W3S mutant, which has an impaired lipid-2 binding site[13] (Figs. S1C, D, S4, S9–S11). The structures were determined at 3.12 Å and 2.35 Å resolution, respectively, by imposing D6 symmetry, although the same reconstruction at a slightly lower resolution was obtained in C1 symmetry (Fig. S5).

The Cx32 GJC structure in nanodisc without addition of CHS during purification confirmed that the density corresponding to lipid-2 is indeed CHS, and this molecule is a prerequisite for stabilizing the NO-state (Fig. 4A, B). Removal of CHS from the purification procedure, followed by nanodisc reconstitution of Cx32, results in a 3D reconstruction where NTH tilts and aligns along the channel pore, displacing the lipid-2 density (Fig. 4B). This conformation is compatible with an open state of a GJC, similar to those observed for Cx26[27] (Fig. 4D, Fig. S12).

## W3S GJCs do not form an NO-state in lipidic environment

Mutation of the residue W3 to a serine produced a similar effect on Cx32 GJCs upon lipid reconstitution in the presence of CHS, as removing CHS did on the wild-type Cx32 (Fig. 4C, Fig. S13A). The NTH of W3S GJC in POPC is ordered and points towards the inside of the

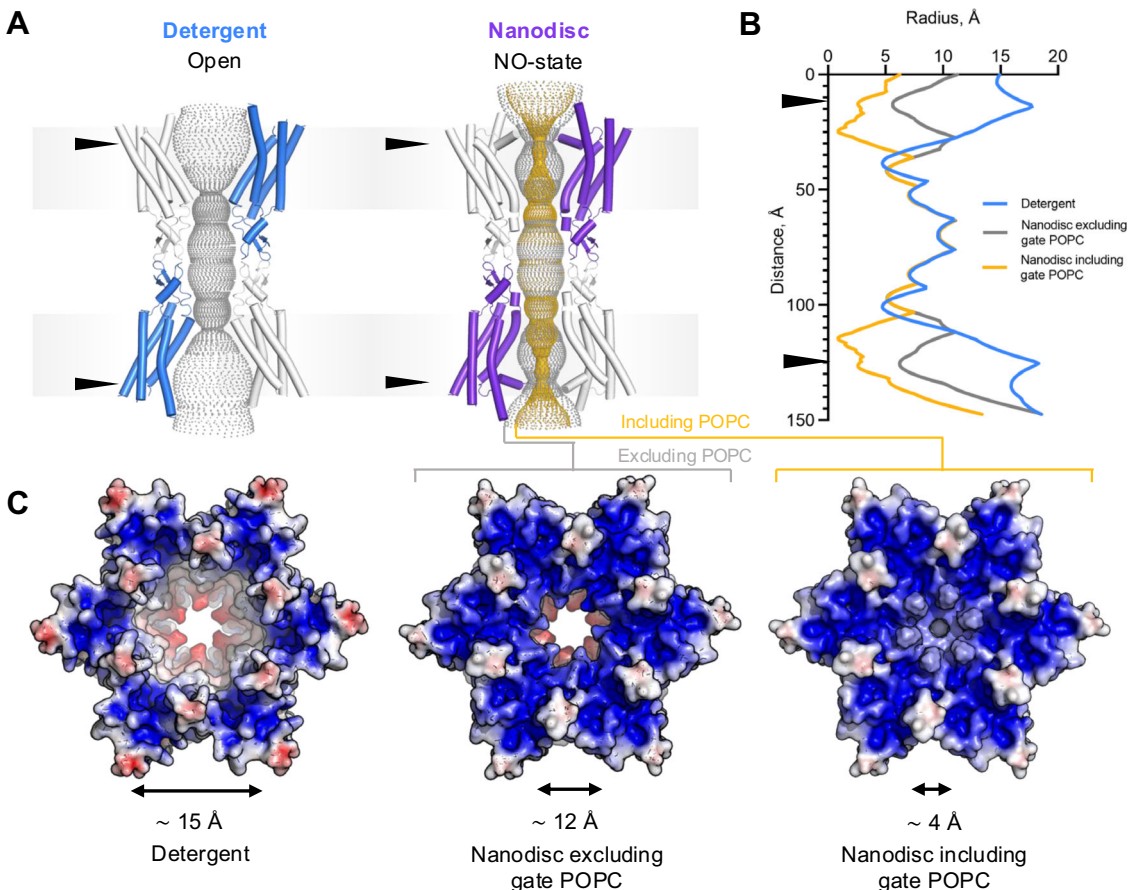

**Fig. 3 | Lipids induce changes in Cx32 GJC surface electrostatic potential and pore diameter. A** Pore representations of Cx32 GJC solved in detergent (PDB ID: 7ZXM[13]) and Cx32 GJC in nanodisc, calculated using the HOLE method. Gray contour represents the pore diameter without and yellow with POPC in the HOLE calculation. The arrows indicate the position of change in the pore radius due to NTH rearrangement. NO-state – ordered N-terminus state. **B** The pore radius of Cx32 GJC, represented in (**A**), calculated using HOLE. Both nanodisc plots represent the POPC nanodisc conditions, including or excluding POPC in the calculation. **C** Electrostatic surface potential of Cx32 GJC, solved in detergent, in nanodisc, and in nanodisc including POPC in electrostatic surface calculation.

channel pore, in a conformation slightly different from that in the wild-type Cx32 maintaining a pore diameter of >10 Å consistent with an open channel (Figs. 4D–F, S12). The conformation of the NTH in the W3S GJC in lipidic environment is distinct from the constricted NTH conformation observed in the Cx32-W3S HC (Fig. S13B)[13]. It is important to note that while structurally the NTH conformations of the W3S GJC in POPC nanodiscs and W3S HC in detergent are distinct, the two states of the mutated protein are also dramatically distinct in their functional properties. Whereas the GJC function of the W3S mutant appears to be very similar to that of the wild-type Cx32, the HC activity of the W3S is dramatically reduced[13]. The current limitations of our protein preparations prohibit HC structure determination, and we are at loss to explain the reason why Cx32HCs are refractory to structure determination upon lipid reconstitution. The optimal approach to correlate the structural and functional properties of W3S GJCs and HCs would be to compare the corresponding structures determined for the proteins reconstituted in the same lipidic environment. Stabilization of wild-type and W3S mutant Cx32 HCs in nanodiscs for cryo-EM structure determination will require careful future experimentation.

To determine whether there is a variability in the NTH conformation among the individual subunits of the W3S and the wild-type Cx32 (in the absence of CHS) GJCs, and whether any of the Cx32 subunits still feature lipid-2 or lipid-3 densities that may be averaged out due by imposing the D6 symmetry, we performed protomer-focused classification (Fig. S7C, D). In all classes for both analyzed datasets we could observe the NTH in only one corresponding major conformation. Neither lipid-2 nor lipid-3 could be

identified in the classes, further confirming that the lipid-2 corresponds to the CHS molecules added during purification.

Altogether, our results with the wild-type Cx32 in the absence of CHS and the Cx32-W3S GJCs indicate the requirement of the sterol binding site (lipid-2 site) for stabilizing the NTH in a conformation that allows the phospholipids to bind to the lipid-3 sites.

## Analysis of Cx32 NTH motions through molecular dynamics simulations

To assess the dynamic interplay between the NTH and the lipids present in the Cx32 pore, we performed molecular dynamics simulations (MD) using a minimal hexameric Cx32 HC system, embedded in a lipid bilayer (see "Methods" for details on the system setup).

For the purposes of our MD analysis, we used cholesterol (CHOL) instead of CHS, because CHOL is a natural sterol that is abundant in biological membranes and is likely to occupy the lipid-2 binding sites in the native Cx32 GJCs/HC. Secondary structure analysis was performed for residues located at the N- and C-termini of each NTH (residues 3−5 and 9−11) in wild-type Cx32 and Cx32 W3S HCs both in the presence and absence of bound lipids, POPC and CHOL. We monitored the average probability of the aforementioned residues adopting either an α-helical or any other type of conformation over the course of two replicate 1 μs MD simulations for each studied system (Fig. 5A, B, E, F). The results indicate a significantly lower probability of these residues forming an α-helix in Cx32 W3S compared to wild-type Cx32 ($p \leq 0.05$, Table S1) (Fig. 5A, B). Interestingly, the disruption of the α-helical secondary structure becomes more significant in Cx32 W3S as the

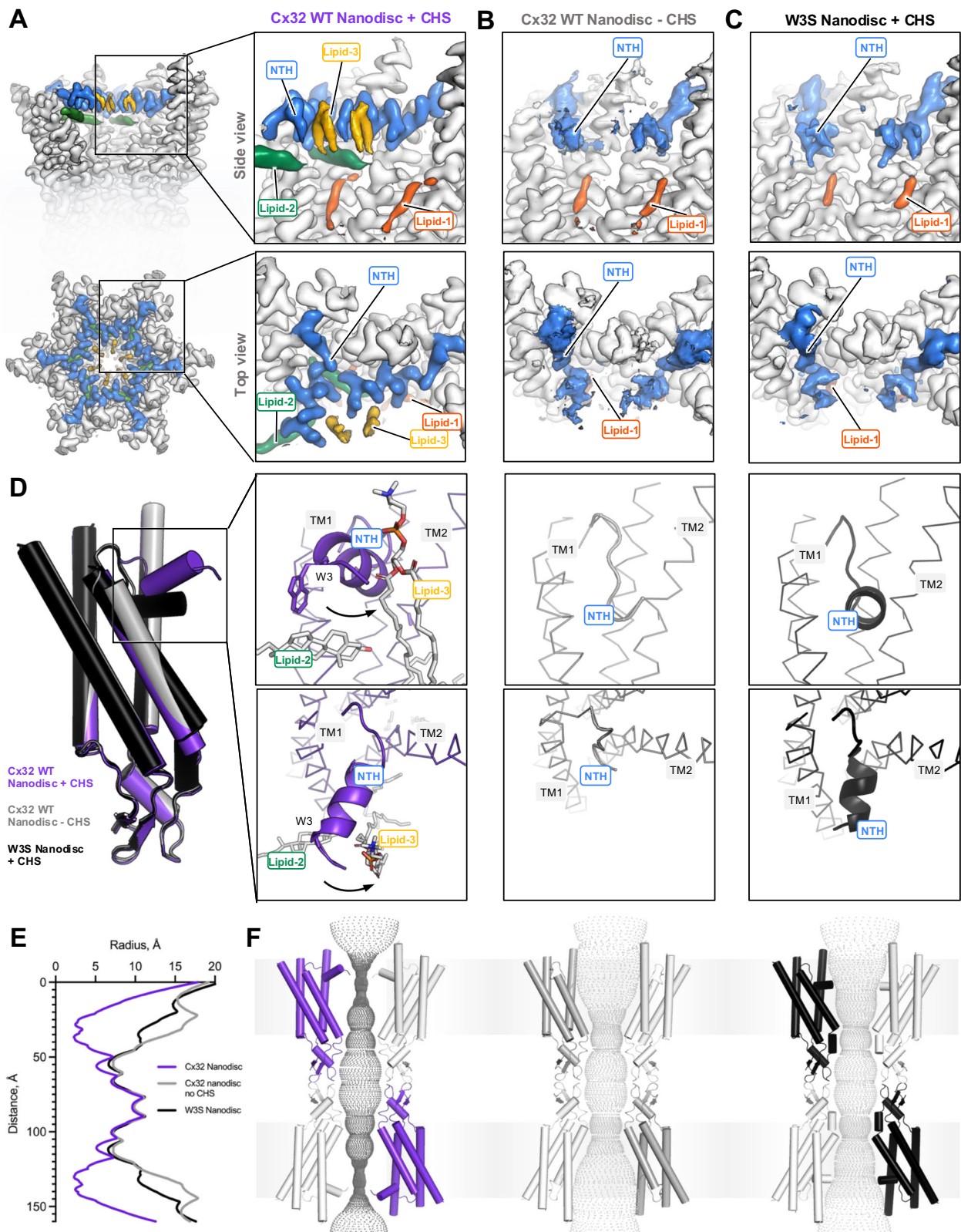

**Fig. 4 | Lipid-2 is necessary for NTH stabilization in NO-state. A** Presence of both lipid-2 and lipid-3 results in an ordered N-terminal helix (NO) confirmation. **B** Removing CHS from Cx32 reconstitution into nanodisc results in a different conformation of the NTH, which appears to be less stable and does not contain a lipid-2 nor lipid-3 density. **C** W3S mutant, which has an impaired lipid-2 binding site, similarly does not have a lipid-2 nor lipid-3 density and contains a less ordered and different NTH conformation from NO-state. **D** The absence of lipid-2 leads to NTH rearrangements, which are different from the NO-state (Cx32 WT Nanodisc + CHS). **E** Graphical representation of pore diameter analysis performed using HOLE, represented in (**F**). **F** Pore diameter analysis of NO-state (Cx32 Nanodisc + CHS), Cx32 WT Nanodisc - CHS, and W3S Nanodisc + CHS, calculated using HOLE software.

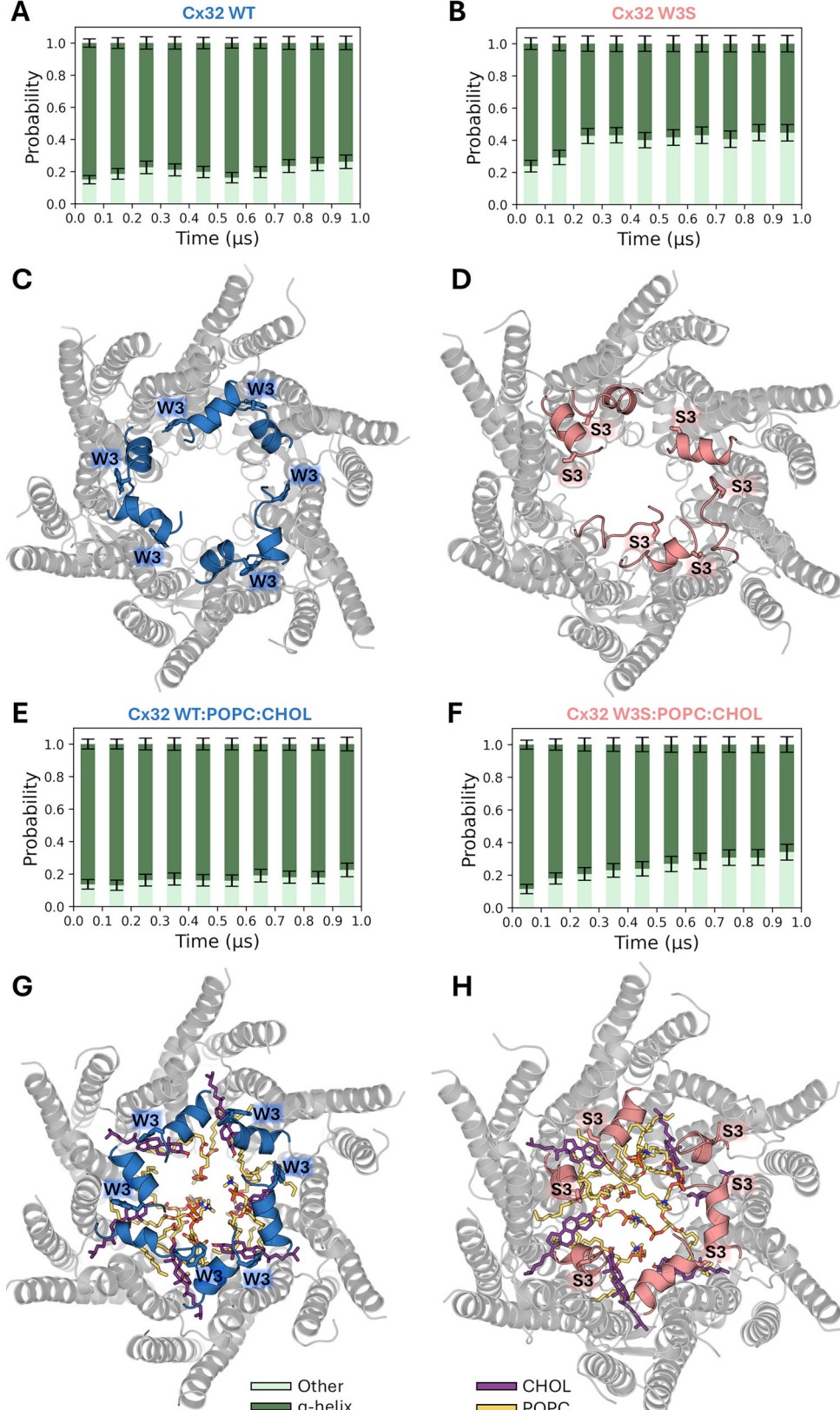

**Fig. 5 | Time evolution of the secondary structure of NTH residues in different Cx32 systems during 1 μs MD simulations. A, B, E, F** The mean probabilities of the NTH residues 3–5 and 9–11 adopting an α-helix or another type of conformation (mainly turns and random coil) for the indicated systems throughout the simulation time. For each system, the plotted mean probabilities were calculated by averaging the per-residue probabilities for the selected residues in every protein chain and over the two replicate MD simulations ($n = 12$) in 0.1 μs intervals (i.e., 0–0.1,..., 0.9–1 μs). Other types of secondary structure are omitted. **C, D, G, H** Structural representation of the last frame ($t = 1$ μs) in one of the two replicate trajectories for each system. The N-terminal residues 1–10 are colored in blue and salmon on each panel. Statistical analysis is presented in Table S1.

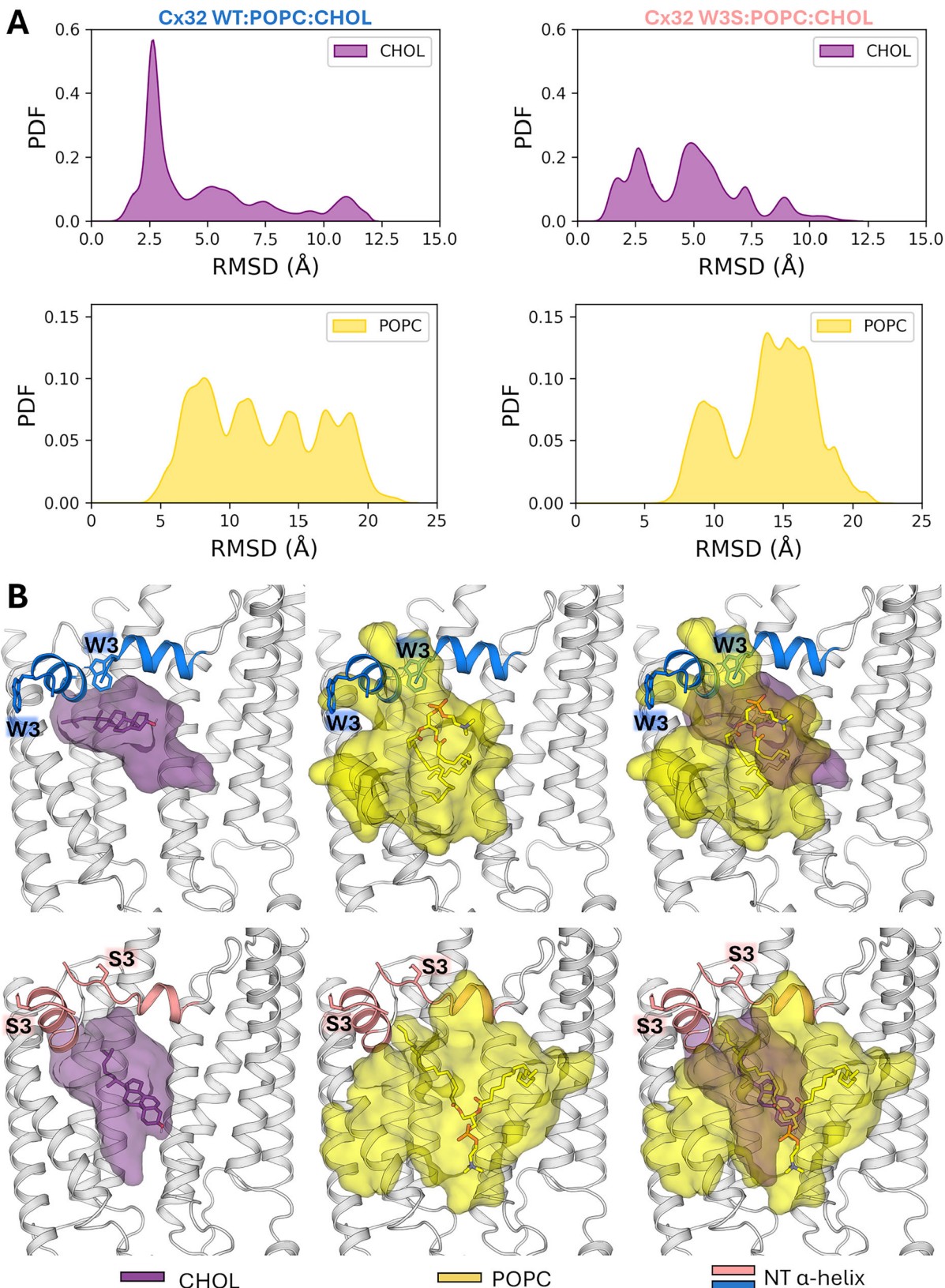

simulation time increases (Fig. 5B). This underscores the instability of the NTHs carrying the W3S mutation, which were initially modeled by retaining the Cβ orientation of W3 in the wild-type protein. On the other hand, the presence of bound POPC and CHOL is likely to enhance the stability of the NTHs in both wild-type Cx32 (Fig. 5E) and Cx32 W3S (Fig. 5F) relative to the respective unliganded forms (Fig. 5A, B),

although this trend did not reach statistical significance ($p > 0.05$, Table S1).

Structural representations of the final frames collected for each analyzed system in one of the replicate 1 μs trajectories are provided to visualize the conformational changes undergone by the NTH residues during the MD simulations (Fig. 5C, D, G, H). The depicted structure of

**Fig. 6 | Dynamic behavior of lipids within the Cx32 pore region during MD simulations. A** Distribution of RMSD values for the heavy atoms of the six CHOL molecules bound to Cx32 in the Cx32 wt:POPC:CHOL and Cx32 W3S:POPC:CHOL complexes. RMSD values were calculated relative to the minimized initial structure, with all trajectory frames from two 1-μs replicate simulations for each system fitted to the protein backbone in the reference structure. The first 100 ns of each trajectory was discarded to ensure equilibrium in the distributions. **B** Structural depictions of the central structures for Cx32 wt:CHOL:POPC and Cx32

W3S:CHOL:POPC, calculated through clustering analysis. Transparent surfaces enclosing the POPC and CHOL molecules were generated after superimposing all protein chains along with the bound lipids within each system. One CHOL molecule and one POPC molecule, located at the centers of their respective accessible regions, are depicted as sticks, while two adjacent protein chains forming the complete CHOL binding site are shown as cartoons. For clarity, three panels are presented for each system: one depicting each lipid individually and one showing both lipids together.

apo Cx32 W3S hexamer (Fig. 5D) clearly illustrates the tendency of some of its NTHs to unfold during the microsecond-long MD simulation. These results suggest that longer MD simulations may be required to observe the unfolding process of all the six NTHs. To a lesser extent, a similar phenomenon can be observed in the last frame of Cx32 W3S in the presence of bound lipids (Fig. 5H). In contrast, the NTHs of wild-type Cx32 remained largely intact at the end of the MD simulations in both conditions (Fig. 5A, G). The conclusions drawn from the visual inspection of the last frames are consistent with the secondary structure analysis presented above.

We extended the same analysis to four systems consisting of wild-type Cx32 and Cx32 W3S, each bound to either CHOL or POPC (Fig. S14). Our results show that the NTH stability in the wild-type protein bound to POPC is significantly higher than that of Cx32 W3S in the same condition ($p \leq 0.05$, Table S1). A similar trend was observed for the CHOL-bound systems ($p = 0.078$, Table S1). Furthermore, the presence of either CHOL or POPC appears to stabilize the NTHs in both wild-type Cx32 and Cx32 W3S relative to their unliganded forms, although this effect was not statistically significant ($p > 0.05$, Table S1).

The RMSF calculations for the NTHs (Fig. S15) underscore the increased flexibility in the N-terminal residues of Cx32 W3S mutant across all conditions, becoming more significant in the absence of lipids. In particular, the RMSF profile in Fig. S15A suggests that the destabilizing impact of the W3S mutation during the microsecond-long MD simulations is especially evident at both ends of the NTH, where residues exhibit greater flexibility compared to their counterparts in wild-type Cx32. This increased flexibility likely contributes to a progressive shortening of the NTH in the mutant protein, which motivated our prior selection of residues at both termini of the NTH for secondary structure probability analysis (Figs. 5 and S14, and Table S1). Complementary RMSD measurements (Fig. S16) also reveal a greater tendency of the N-terminal residues in the mutated protein to deviate from their initial α-helical conformation compared to those in wild-type Cx32.

To further evaluate the impact of the W3S mutation on the stability of the NTH, we carried out alchemical free energy calculations using the thermodynamic integration (TI) method. A thermodynamic cycle in which the mutation was performed in both in the native conformation of the NTH within the Cx32 HC and in a linear, capped tripeptide model (ACE-Gly-X-Gly-NME, where X is either W or S[28]; enabled the prediction of the folding free energy difference between the wild-type and the mutant proteins (ΔΔG, Eq. S1). Integration of the $\langle \partial V / \partial \lambda \rangle_\lambda$ vs. $\lambda$ profiles for each alchemical transformation, i.e., discharging, van der Waals, and recharging steps (Fig. S17), yielded a ΔΔG of $-3.62 \pm 0.46$ kcal/mol (Table S2). The negative value of the estimated ΔΔG indicates that the folding free energy of Cx32 wt is more favorable than that of Cx32 W3S, consistent with the destabilizing role of the analyzed mutation. The influence of lipids on NTH motion relative to TM1 was investigated by monitoring an angle during MD simulations, defined by the centers of mass of three atom groups at the NTH, the N-terminal end of TM1, and the center of TM1 (see Fig. S18). The angle distributions for wild-type Cx32 show that neighboring NTHs tend to alternate between conformations shifted upward and downward relative to the position observed in the cryo-EM structure of Cx32 in nanodiscs (Fig. S19). A similar trend was observed for Cx32 W3S, with broader angle distributions reflecting partial NTH

unfolding (Fig. S20). In contrast, for wild-type Cx32 and W3S with POPC and CHOL at the lipid binding sites, the angle distributions are narrower, suggesting that the bound lipids restrict the NTH motion to positions closer to that observed in the experimental structure (Figs. S21 and S22).

Overall, the results presented here highlight the destabilizing effect of the W3S mutation on the Cx32 NTH. Moreover, we observed that the binding of CHOL and/or POPC is likely to exert a stabilizing influence on these helices in both the mutant and wild-type proteins. Disruption of the CHOL binding site due to the unfolding of the NTH regions in the W3S mutant makes the presence of ordered lipids within the pore less likely based on our cryo-EM data.

## Dynamic behavior of lipids within the Cx32 pore region

To investigate the conformational dynamics of lipids bound to wild-type Cx32 and Cx32 W3S during the 1-μs replicate MD simulations, we tracked the RMSD values for their heavy atoms relative to the corresponding starting conformations in the initial structure (Fig. 6A–H). The distribution of RMSD values for CHOL indicates that these molecules remained largely in native-like conformations when bound to the wild-type protein, as evidenced by a high and narrow peak around ~2.5 Å (Fig. 6B). In contrast, a wider RMSD distribution was observed for CHOL bound to Cx32 W3S (Fig. 6B), likely due to partial disruption of the NTH residues in the mutated system (Fig. 5F, H). This structural change may compromise the integrity of the CHOL binding site.

In both systems, POPC molecules exhibit broad RMSD distributions throughout the simulations, reflecting their high intrinsic flexibility and binding to a relatively shallow surface on the protein (Fig. 6A). Notably, the main peak of RMSD values for the POPC molecules bound to Cx32 W3S is shifted toward higher values compared to the wild-type system's distribution. This suggests that the mutation can also compromise the binding of POPC to the NTH region of Cx32.

Figure 6B shows the central structures of wild-type Cx32 and Cx32 W3S bound to POPC and CHOL, determined through clustering analysis. Transparent surfaces were generated to enclose the conformational space explored by the ligands' central structures after superposition of all chains within each system. These structures highlight the loss of the native binding mode of CHOL to Cx32 W3S and reveal a tendency for POPC molecules to adopt conformations in the mutant that are more detached from the NTH compared to the wild-type system. Thus, the MD simulations suggest that CHOL and POPC exhibit stronger binding to wild-type Cx32 than to Cx32 W3S, consistent with experimental findings.

## Dominant motions of wild-type Cx32 monomers potentially enabling lipid entry into the pore

Despite mounting evidence supporting the presence of lipids into the Cx32 pore, the entry path of the external lipids remains unknown. To shed some light into this issue, we assessed the dominant motion of the Cα atoms of every Cx32 monomer across all the simulated wild-type HCs using principal component analysis (PCA). Our results indicate that the two largest motions, corresponding to the principal components 1 and 2 (PC1 and PC2), account for nearly 50% of the total protein motion (Fig. 7A). By projecting the sampled phase space on these two PCs, we observed different regions (Fig. 7B), characterized

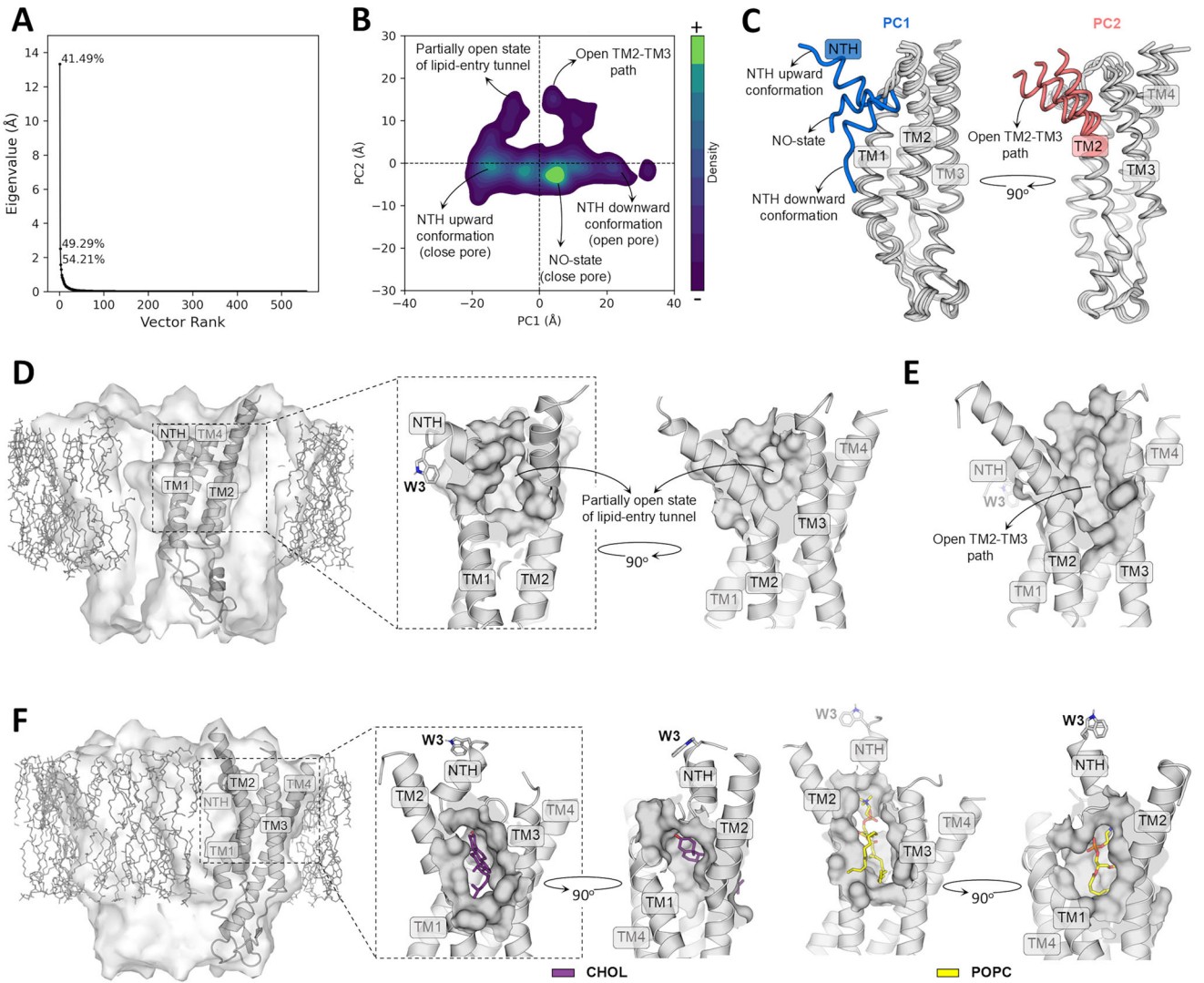

**Fig. 7 | Dominant motions of Cx32 wt monomers during hemichannel MD simulations and their implications for bilayer lipid entry into the pore. A** PCA eigenvalue spectrum showing the contribution of each PC to the motion of the Cx32 wt Cα atoms. The cumulative variance explained by the top three PCs is annotated on the graph. **B** Density plot of Cx32 wt motion projected onto the top two PCs. Regions of the phase space corresponding to structural features of interest are indicated. **C** Superimposed structures extracted along PC1 and PC2, highlighting the extent of motion captured by the principal components. **D** Structural representation of a Cx32 wt monomer extracted from a conventional MD trajectory frame, bearing a partially open tunnel connecting the hemichannel's pore to the lipid bilayer (hypothesized to act as a lipid-entry path). A surface

representation of the entire hemichannel embedded in the lipid bilayer (gray sticks), with a Cx32 wt monomer shown in cartoon, is provided to contextualize the monomer's orientation in the zoomed-in views of the open cavity. **E** Structural representation of a Cx32 wt monomer extracted from a conventional MD trajectory frame, exhibiting a wide TM2–TM3 opening that could allow the passage of lipids from the bilayer into the pore. In this structure, the passage through TM1–TM2 is blocked by NTH. **F** Structural depictions of two trajectory frames from steered MD simulations, initiated from the Cx32 wt conformation shown in (**E**), during which cholesterol and POPC molecules were pulled from the bilayer into the pore and the NTH was concomitantly moved upward to open the TM1–TM2 path.

by structural properties that could be relevant to understand the lipid entry, as we will be described hereinafter.

To characterize the dominant motions of wild-type Cx32 monomers, we extracted conformations along PC1 and PC2 corresponding to the minimum, mean, and maximum values of each PC. Structural representations of these frames show that PC1 describes the upward and downward displacement of the NTH along the pore axis, whereas PC2 reflects the lateral displacement of TM2 relative to TM3 (Fig. 7C). Notably, the motion associated with PC1 encompasses the previously described open (downward conformation) and NO states, as well as the identified upward conformation (Fig. 7B, C). These findings are consistent with earlier results derived from NTH–TM1 angle distributions of wild-type Cx32 (Fig. S19). Given that the displacements of the NTH and TM2 represent the most prominent dynamic modes of wild-

type Cx32, they are hypothesized to contribute to lipid translocation from the bilayer into the pore.

Interestingly, a wild-type Cx32 monomer frame sampled from a phase space region characterized by negative PC1 values and large positive PC2 values (Fig. 7B) exhibits a partially open tunnel connecting the bilayer with the pore interior (Fig. 7C). However, to enable the passage of lipids such as POPC and cholesterol, this tunnel would require a wider opening, especially at the TM2–TM3 interface. Another monomer, extracted from a region with positive PC1 and a PC2 value higher than that of the previous structure (Fig. 7C), possesses a substantially widened TM2–TM3 opening that could, in principle, allow lipid passage (Fig. 7E). However, in this conformation, the NTH occludes the exit into the pore. These structural analyses suggest that a combination of a large displacement of TM2

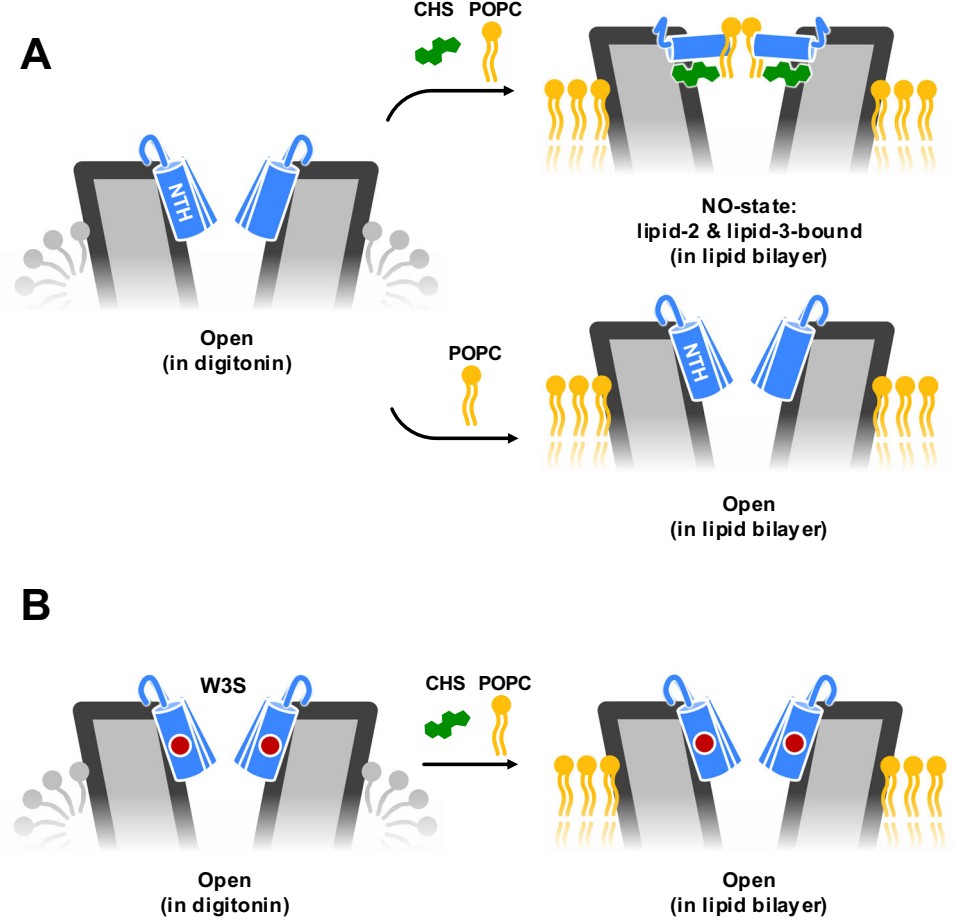

**Fig. 8 | A schematic representation of the N-terminus changes in Cx32 GJC in response to changes in lipid composition. A** Structures of Cx32 GJCs determined using protein in detergent micelles feature a dramatically different conformation of the NTH, compared to those determined in lipid bilayer (in nanodiscs). The observed conformations depend critically on the presence of sterols (CHS). The NTH is represented in blue, the sterol molecules in green, and phospholipids in yellow. **B** The CMT1X disease-linked mutation W3S is insensitive to the sterol effects on the NTH structure. These cryo-EM observations are consistent with the MD simulations, which reveal a mutually stabilizing role of the lipids and the NTH residues.

away from TM3, coupled with an upward movement of the NTH, would be necessary to enable lipid entry into the pore. Such a concerted motion was not fully captured in the MD simulations conducted here, likely reflecting the inability of unbiased MD to efficiently sample phase space regions separated by high energy barriers. It is also possible that the insertion of lipids into the TM2–TM3 opening, a phenomenon not observed in the analyzed trajectories, could facilitate the upward displacement of the NTH thereby contributing to the formation of the TM1–TM2 exit pathway.

To further evaluate the feasibility of lipid translocation into the pore via the TM2–TM3 and TM1–TM2 openings, we performed steered molecular dynamics (SMD) simulations in which a cholesterol and a POPC molecule, positioned near the TM2–TM3 path entrance, were independently pulled toward the pore. These simulations were initiated from a trajectory frame displaying a fully open TM2–TM3 pathway, as shown in Fig. 7E. Prior to lipid pulling, the NTH–TM1 angle was increased to ~100° to fully open the initially closed TM1–TM2 pathway. Frames extracted from lipid-pulling simulations suggest that concerted movements of the NTH and TM2 can create a tunnel sufficiently wide to allow lipid translocation into the pore (Fig. 7F). However, the conclusions drawn from this approach remain preliminary and constitute an extrapolation of the dominant motions observed during the MD simulations of wild-type Cx32 HCs. A further limitation arises from the absence of the TM2–TM3 connecting segment in the simulated system, as this region is unresolved in the cryo-EM structure. Its

presence could restrict TM2–TM3 motion, potentially rendering lipid translocation through this pathway less likely.

## Discussion

The structures of Cx32 GJC in different lipidic environments provide a mechanistic insight to the previously suggested effects of lipid composition on Cx32 channel function[5–7] and NTH-mediated gating[13]. The NTH is a dynamic element of the connexin channel structure, reacting to the presence of different lipid species by changing its conformation and thus regulating the solute permeation through the Cx32 GJCs (Fig. 8). In the presence of phospholipids (POPC or LPL) the NTH reorient from lining the channel pore (observed in the open conformation) to pointing toward the channel pore axis. However, binding of sterols to the lipid-2 site is required for this. The switch from open to NO-state is associated with changes in both the electrostatic properties in the radius of the pore, suggesting that these NTH transitions change the range of molecules that can pass through the channel.

It is important to note that our MD simulations reveal a layer of complexity in lipid-protein interactions that is not apparent when observing the "static" cryo-EM structures. Our cryo-EM reconstruction of the Cx32 GJC in lipidic environment indicates that a conformation of the NTH is formed whereby phospholipids and sterols work together to lock the protein in a particular conformation. In stark contrast, our MD simulations reveal a great degree of plasticity of the lipid binding sites within the pore, especially the putative

phospholipid interactions sites formed by the NTH regions, as suggested by the cryo-EM reconstructions. It remains to be determined whether additional factors play a role in stabilizing both the lipids and the Cx32 NTH residues.

We have observed previously that the CMT1X-causing Cx32 mutant W3S has a closed NTH conformation in the cryo-EM structures of W3S HC in detergent, correlating with the functional effect on the W3S HC function[13]. Interestingly, the Cx32 GJCs appeared unaffected by the W3S mutation. The results presented here suggest that reconstituting Cx32 into a lipidic environment allows us to sample the phospholipid-induced conformational changes of the NTH: while wild-type Cx32 is capable of interacting with POPC (or other phospholipids present in the LPL mixture) via the lipid-3 site in the lipid-2-dependent manner, the W3S mutant fails to establish the lipid-2 site and thus does not form a lipid-3-bound NO-state.

Previous structural studies on Cx43 GJC[10,11], Cx36 GJC[12], and Cx32 HC[13] have implicated the NTH conformational changes in regulation of the connexin channel permeability and suggested a link between these conformational changes and distinct lipid environments. For example, a comprehensive structural analysis of the conformational changes in Cx43 revealed the existence of distinct conformation states of the NTH: the gate-covering, pore lining, and flexible intermediate NTH states[10]. Moreover, Lee et al. found that the presence of CHS, varied pH and use of a C-terminally truncated Cx43 construct can influence the NTH conformation[10]. It remains to be determined whether phospholipids and sterols bind and induce the observed NTH rearrangements in a physiological context. It is tempting to suggest that localization of Cx32 in cholesterol-rich membrane microdomains[8] may predispose Cx32 to interactions with cholesterol via various sites at the outer or inner surface of the channel. However, the path taken by either cholesterol or the phospholipids to reach the lipid-2 or lipid-3 sites, respectively, is not immediately obvious, as these discrete sites are separated from the lipid bilayer by the protein. Delineation of the route used by the lipids to dynamically regulate Cx32 channels via lipid-2 and lipid-3 sites will require careful experimental and computational analysis.

It is worth noting that gating by lipids has become a topic of intensive investigations not only in connexins, but also in other members of the large pore channel family. For example, obstruction of the pore by phospholipids has been observed in the *Caenorhabditis elegans* innexin-6 channels reconstituted into nanodiscs[29]. The volume-regulated LRRC8A:C channels have been shown to utilize the intra-pore lipids to block ion conduction[30]. Another member of the family, the calcium homeostasis modulator 1, CALHM1, has been shown to accommodate sterols and phospholipids in a conserved lipid-binding pocket; in the case of CALHM1 phospholipid binding appears to stabilize and regulate the channel[31,32]. Lipid-mediated regulation has emerged as a common denominator in those studies. We anticipate that future investigations probing the molecular mechanisms of the large pore ion channel family members will reveal the common mechanistic features of the lipid-mediated modulation of these proteins' structure and function, which may reflect our own observations on lipid-reconstituted Cx32 channels.

## Methods

### Lipid preparation
The lipids (1-palmitoyl-2-oleoyl-glycero-3-phosphocholine (POPC) or bovine liver polar lipid extract (LPL)) (Avanti Polar Lipids) dissolved in chloroform at a concentration of 25 mg/ml were transferred to a glass tube and dried under the nitrogen stream until completely dried. A buffer consisting of 25 mM Tris-HCl pH 8.0, 150 mM NaCl, 2% n-Dodecyl β-D-maltoside (DDM) was added to reach lipid concentration of 8.33 mg/ml. The lipids were placed in a sonication bath at 45 °C until the lipids were completely dissolved; the clarified lipid stocks were aliquoted and stored at −20 °C until use.

### MSP2N2 expression and purification
A single colony of *E. coli* BL21(DE3) cells transformed with MSP2N2 plasmid was used to inoculate 100 ml of LB medium supplemented with 50 μg/ml kanamycin and grown overnight at 37 °C and 180 rpm. Terrific broth (TB) medium (1 L), supplemented with 50 μg/ml kanamycin, was inoculated with 30 ml of the overnight culture, and grown at 37 °C and 180 rpm to the $OD_{600}$ of ~3.0. Protein expression was induced by addition of IPTG at a final concentration of 1 mM for 3 h at 37 °C and 180 rpm. Cells were harvested by centrifugation for 30 min at $3700 \times g$ and 4 °C. The cell pellets were resuspended in 50 ml of 40 mM Tris-HCl pH 8.0, 0.3 M NaCl and 1% Triton X-100 per 1 L of cell culture, supplemented with 1 mM PMSF and 10 μg/ml DNase I and lysed by sonication for 15 min on ice at 50% amplitude. The lysate was clarified by centrifugation for 30 min at $20,000 \times g$. The cleared lysate was added to Ni-NTA resin (2 ml per 1 L of cell culture) and incubated for 30 min at 3 °C with constant rotation. The resin was collected in a gravity column, washed with 20 column volumes (CV) of 40 mM Tris-HCl pH 8.0, 0.3 M NaCl, 50 mM Na-cholate and 20 mM imidazole, then 20 CV of the same buffer with 50 mM imidazole and eluted with 5 CV of the same buffer with 400 mM imidazole. The protein was desalted into 20 mM Tris-HCl pH 8.0, 0.1 M NaCl, 0.5 mM EDTA using G-25 PD-10 desalting column, concentrated using an Amicon Ultra 10 kDa cut-off concentrator, aliquoted and stored at −80 °C until use.

### Cell culture
HEK293F cells were maintained in 15 cm plates in Dulbecco's modified Eagle medium (DMEM) supplemented with 10% fetal calf serum (FCS) and 1% Penicillin-Streptomycin (PenStrep) at 37 °C and 5% $CO_2$. For transfection using branched polyethyleneimine (PEI), the medium was replaced with DMEM supplemented with 2% FSC PenStrep. The DNA and PEI dilutions were prepared separately in un-supplemented DMEM, using 40 μg DNA per plate and PEI in 1:2 ratio of DNA to PEI (w/w). The dilutions were mixed and incubated at room temperature for 5 min and added to the cells in the drop-wise manner. The cells were cultured at 37 °C and 5 % $CO_2$ for 48 h, after which they were harvested by scraping and stored at −80 °C.

### Connexin purification
The protein purification was performed as described previously[13]. The adherent HEK293F cells were transfected with a pACMV plasmid encoding Cx32 or Cx32-W3S mutant, in frame with a C-terminal 3C-YFP-twinStrep tag. Following a 48 h period after transfection, cells harvested from 100 15-cm culture plates were resuspended in buffer A (25 mM Tris-HCl pH 8.0, 150 mM NaCl) supplemented with protease inhibitors (1 mM benzamidine, 1 μg/ml leupeptin, 1 μg/ml aprotinin, 1 μg/ml pepstatin, 1 μg/ml trypsin inhibitor and 1 mM PMSF) and 10 μg/ml DNase I. The cells were lysed by 300 sonication pulses at 35% amplitude (Sonics Vibra-Cell, cycle: 0.5 s pulse on, 0.5 s pulse off). The lysate was clarified by ultracentrifugation at 4 °C, $142,000 \times g$ (Ti45 rotor, Beckmann Coulter) for 50 min. The pellet was transferred to fresh buffer A supplemented with protease inhibitors and homogenized. The membranes were solubilized by addition of 1% DDM and 0.2% cholesteryl hemisuccinate (CHS) for 1 h at 4 °C, with constant rotation. In the case of purifying Cx32 in the absence of CHS, CHS was omitted in this membrane solubilization step. The membranes were clarified by ultracentrifugation at 4 °C, $142,000 \times g$ (Ti45 rotor) for 50 min. The solubilized and clarified membranes were incubated with 2 ml of anti-GFP nanobody coupled CNBr-sepharose resin for 30 min at 4 °C, with constant rotation. The resin was applied to a gravity column, washed with 40 column volumes of buffer C (25 mM Tris-HCl pH 8.0, 150 mM NaCl, 0.1 % digitonin). The protein was eluted by cleavage using 3 C protease (0.4 mg) for 2 h with rotation at 4 °C. The eluted protein was concentrated to a volume <800 μl using AmiconUltra 100 kDa cut-off concentrator.

## Nanodisc reconstitution

For nanodisc reconstitution, the purified Cx32 protein was mixed with MSP2N2 and lipids, either 1-palmitoyl-2-oleoyl-glycero-3-phosphocholine (POPC) or liver polar lipids (LPL), diluted in buffer C to 200 μl, in a molar ratio of 6:2:200 (Cx32:MSP2N2:lipid) and incubated at room temperature at constant rotation for 1 h. The Bio-Beads SM2 Resin (BioRad, 50 mg) was added to the mixture, placed at 4 °C with constant rotation overnight. The protein was separated from the BioBeads into a fresh Eppendorf tube, centrifuged for 5 min at 24,000 × *g* and 4 °C using a cooling benchtop centrifuge (Eppendorf). The reconstituted protein was further purified by HPLC using a Superose 6 Increase 10/300 GL column pre-equilibrated with buffer A. The size exclusion chromatography (SEC) fractions corresponding to Cx32 in nanodisc were pooled together and concentrated to ~2 mg/ml (samples: Cx32 POPC nanodisc, Cx32 LPL nanodisc, W3S POPC nanodisc) or ~1.3 mg/ml (sample: Cx32 noCHS POPC nanodisc) for cryo-EM sample preparation.

## Cryo-EM sample preparation and data collection

For cryo-EM sample preparation, 3.5 μl aliquots of concentrated protein reconstituted in nanodiscs were applied to glow discharged Quantifoil Cu R1.2/1.3 200-mesh grid. The grids were blotted for 3 s and plunge-frozen in liquid ethane using Vitrobot Mark IV (Thermo Fisher).

The cryo-EM datasets were collected using a Titan Krios electron microscope (Thermo Fisher), equipped with a K3 direct electron detector camera (Gatan) and a GIF-Quantum energy filter, with a slit width of 20 eV. The defocus range was set from −0.5 μm to −2.5 μm. The data was collected as movies dose-fractionated into 40 frames using EPU 2.0. The exposure time for each micrograph was 0.9 s, with a total dose of 61.6 e⁻/Å² for Cx32 POPC nanodisc dataset, 61.2 e⁻/Å² and 50 e⁻/Å² for Cx32 LPL nanodisc datasets, 55 e⁻/Å² for Cx32 without CHS POPC nanodisc dataset, and 55 e⁻/Å² for W3S POPC nanodisc dataset.

## Cryo-EM data processing

Optics groups were assigned to individual movies using a script provided by Dr. Pavel Afanasyev[33]. The movies were motion-corrected using MotionCor2[34]. The Cx32 POPC nanodisc and Cx32 LPL datasets were CTF-corrected using Gctf[35], and Cx32 no CHS POPC nanodisc, and W3S POPC nanodisc datasets using CTFFind4[36]. The two Cx32 LPL nanodisc datasets were merged immediately after CTF refinement. Approximately 1000 particles were picked per dataset, extracted, and subjected to one round of 2D classification in Relion 4.0.1[37]. The best classes, showing features of GJC and HC, were selected for reference-based autopicking in Relion 4.0.1. The autopicked particles were subjected for several rounds of 2D classification, until clear GJC features were observed. Best particles were selected for 3D classification, using Cx32 GJC in detergent as a reference (low pass filtered to 40 Å)[13], in D6 symmetry. The best class, showing clear GJC features, was then subjected to 3D auto-refinement with imposed symmetry followed by CTF refinement and particle polishing and final refinement. Autorefinement without imposing any symmetry (C1) was used to assess the influence of symmetry on map quality and features. The local resolution was determined using ResMap[38]. The detailed steps of the processing pipeline for each sample are described in detail in Figs. S2–S3, S7, S9–10 and Table 1.

Protomer-focused classification was performed by symmetry expanding the particles of the final autorefinement job, using 'relion_particle_symmetry_expand' command, using D6 symmetry. The reference map for a single protomer was generated in ChimeraX[39], based on the corresponding model, using the 'Color Zone' and 'Split Map' commands. The reference was used for mask generation in Relion 4.0.1, with 10 Å low pass filtering, extending the binary map with 5 pixels and adding a 20-pixel soft edge. The particles were then subtracted based on the generated mask, recentred on the mask, and rescaled to box size of 200 × 200 pixels. The particles were used for 3D classification without image alignment and symmetry imposition, using appropriately recentred and rescaled mask and reference, to 4, 6 or 8 classes. The best class number was selected by evaluating if more classes reveal more information in the regions of interest without concomitant drastic decrease in resolution (Fig. S7).

## Model building and refinement

The models of Cx32 GJC in POPC nanodisc, Cx32 without CHS GJC in POPC GJC nanodisc, and Cx32 GJC in LPL nanodisc were built based on the single subunit of Cx32 WT HC detergent model (PDB ID: 7ZXN). The models of W3S GJCC in POPC nanodisc and R22G GJC in POPC nanodisc were built based on the single subunit of W3S and R22G HC detergent models, respectively (PDB ID: 7ZXT and 7ZXO). The models were built in COOT[40] by aligning the subunit template into the density and refining amino acid positions. The cytoplasmic regions (CL: L106-H126 (Cx32 POPC, Cx32 LPL, W3S POPC); CT: A218-C283 (Cx32 POPC, Cx32 LPL, W3S POPC) were not built due to not being resolved in the density map. The cholesterol (lipid2) (chemical ID: CLR) was built into Cx32 POPC nanodisc maps, based on Cx32 HC detergent model (PDB ID: 7ZXN). POPC molecule (chemical ID: POV) was built into Cx32 POPC and LPL nanodisc maps, using first rigid body fit and later real space refine functions in COOT. The side-chain clashes and overlaps were minimized using Chiron[41]. All models were refined using 'phenix.real_space_refine' function and validated using MolProbity[42] in PHENIX[43]. The HOLE analysis of the pore radius was performed in COOT.

## Electrostatic surface potential analysis

The electrostatic potential was calculated in PyMOL using APBS Tools 2.1, using AMBER99 force field.

## System setup for molecular dynamics simulations

The coordinates for a Cx32 wt HC bound to six POPC and six CHOL (Cx32 wt:POPC:CHOL) molecules were extracted from the cryo-EM structure. The protonation states of the protein's ionizable residues and the transmembrane region were predicted using the H++ and PPM web servers, respectively[44,45]. ACE and NME caps were added to residues M105 and I127, respectively, in each Cx32 chain, corresponding to gaps found in the cryo-EM structures.

CHARMM-GUI[46] was employed to embed the protein into a lipid bilayer containing major components of plasma membranes from mammalian cells, i.e., 1-palmitoyl-2-oleoylphosphatidylcholine (POPC), 1-palmitoyl-2-oleoylphosphatidylethanolamine (POPE), palmitoyl sphingomyelin (PSM), 1-palmitoyl-2-oleoylphosphatidylserine (POPS) and cholesterol (CHOL), that are parametrized in the lipid21 force field of Amber 22 (https://opm.phar.umich.edu/biological_membranes/lipid_composition). The resulting bilayer comprised a total of 146 CHOL molecules, 126 POPC molecules, 68 PSM molecules, 68 POPE molecules and 22 POPS molecules. Water molecules were added to form a cuboid box measuring 134.37 Å × 135.51 Å × 102.23 Å, with edges positioned at least 10 Å away from the protein surface and the bilayer stretching along the xy plane. Table 2 summarizes the composition and box dimensions of all simulated systems.

The system's PDB file, assembled using CHARMM-GUI, was processed by the program charmmlipid2amber.py from AmberTools 22 in order to convert the CHARMM naming convention into AMBER[47]. The PDB files for other systems prepared for MD simulations, which included Cx32 wt with no lipids in the pore (apo Cx32 wt), apo Cx32 W3S, Cx32 W3S with POPC and CHOL in the pore (Cx32 W3S:POPC:CHOL), Cx32 wt with only POPC in the pore (Cx32 wt:POPC), Cx32 wt with only CHOL in the pore (Cx32 wt:CHOL), Cx32 W3S with only POPC in the pore (Cx32 W3S:POPC) and Cx32 W3S with only CHOL in the pore (Cx32 wt:CHOL), were generated from the processed PDB of Cx32 wt:POPC:CHOL, either the initial structure or an equilibrated frame

**Table 2 | Description of composition and simulation boxes for systems prepared for MD simulations**

| System | No. of counterions | Bilayer lipid composition | No. of water molecules | No. of atoms | Initial box dimensions (Å) |
|---|---|---|---|---|---|
| Cx32 WT | 4 Na⁺ | 146 CHOL: 126 POPC:68 PSM: 68 POPE: 22 POPS | 27,422 | 147,726 | x: 134.37 |
| Cx32 W3S | | | | 147,648 | y: 135.51 |
| Cx32 WT:POPC | | | | 148,530 | z: 102.23 |
| Cx32 W3S:POPC | | | | 148,452 | |
| Cx32 WT:CHOL | | | | 148,170 | |
| Cx32 W3S:CHOL | | | | 148,092 | |
| Cx32 WT:POPC:CHOL | | | | 148,974 | |
| Cx32 W3S:POPC:CHOL | | | | 148,896 | |

(see below). In all cases, "CONECT" records for S-S bonds in the proteins were obtained using *pdb4amber* of Amber 22[47], which ensured the correct formation of such bonds before further processing for all-atom MD simulations.

The program *tleap* was employed to obtain the topologies and the coordinate files for the studied systems and to add the counterions (4 Na⁺, Table 2) necessary to neutralize the simulation boxes. The force fields ff19SB and lipid21 were chosen to parametrize the protein and the lipids, including those in the bilayer or within the protein pore. The water molecules were treated with the OPC model[47]. Hydrogen mass repartitioning was carried out to increase the time step from 2 to 4 fs during the production runs[48]. It is worth noting that the initial structures to generate the topology and coordinate files of systems involving the W3S mutation were derived from the equilibrated structures (see below) of the wild-type systems, by manually renaming the W3 residue to SER in the PDB files and deleting the side-chain atoms beyond the CB. The missing hydroxyl group of the Ser residue was added automatically by *tleap*.

**Molecular dynamics simulations**
Prior to the production runs, all systems were subjected to energy minimization (EM) and equilibration steps. A total of 3000 EM cycles were performed for each system using the program *pmemd.MPI*[47]. The minimized systems were subjected to heating in two subsequent simulations. During the first phase, the temperature was linearly increased from 10 to 100 K during 500 ps. Random initial velocities drawn from a Maxwell-Boltzmann distribution at 10 K were assigned to every atom at the beginning of this heating process. Then, during the second phase, the temperature was linearly increased from 100 to 303 K during 600 ps. In both cases, the heavy atoms of the protein and all lipids were restrained to their initial positions using a harmonic potential defined by a spring constant of 10 kcal·Å⁻²·mol⁻¹. The cell volume was kept constant and the temperature was controlled using the Langevin thermostat, with a collision frequency of 1.0 ps⁻¹.

Following the heating phase, the systems underwent four NPT equilibration steps, during which position restraints were applied exclusively to the heavy atoms of the protein and the lipids within the pore (if present). The restraint constant was reduced from 8 to 2 kcal·Å⁻²·mol⁻¹ in 2 kcal·Å⁻²·mol⁻¹ strides over the four 8 ns MD simulations. During these equilibration steps, the systems were simulated at constant temperature and pressure of 303 K and 1 bar, which was achieved by using the Langevin thermostat and the Monte Carlo barostat, respectively. Finally, all systems were subjected to 1 μs production runs in the NPT ensemble, in identical conditions to those used during the NPT equilibration except for the position restraints, which were eliminated. All the steps, from heating to production runs, were carried out in duplicate with *pmemd.cuda* of Amber22[49]. Short-range electrostatic and van der Waals interactions were explicitly calculated using a 10 Å cutoff, while long-range electrostatic interactions were treated with the Particle Mesh Ewald (PME) method.

It is worth noting that, as each Cx32 HC contains six monomers, the two replicate MD trajectories per system correspond to twelve independent monomer simulations, providing statistical robustness for the monomer-based analyses.

**Thermodynamic integration free energy calculations**
Thermodynamic integration (TI) free energy calculations were performed to assess the impact of the W3S mutation on the stability of the Cx32 NTH. The folding free energy difference between the wild-type and mutant proteins (ΔΔG) was determined based on a thermodynamic cycle described in detail elsewhere, in which the reference unfolded state is modeled as a capped tripeptide (ACE-GXG-NME), where **X** represents the residue subjected to mutation[28]. According to this cycle, ΔΔG is calculated as (Eq. 1):

$$\Delta\Delta G \equiv \Delta G_{\text{Fold}}^{wt} - \Delta G_{\text{Fold}}^{mut} = \Delta G_{\text{Unfolded}}^{W3 \to S3} - \Delta G_{\text{Folded}}^{W3 \to S3} \quad (1)$$

where $\Delta G_{\text{Fold}}^{wt}$ and $\Delta G_{\text{Fold}}^{mut}$ correspond to the folding free energies of the wild-type and the mutant proteins, respectively. On the right-hand side, $\Delta G_{\text{Unfolded}}^{W3 \to S3}$ and $\Delta G_{\text{Folded}}^{W3 \to S3}$ stand for the free energy changes associated with the alchemical W3S mutation in the unfolded and folded states, respectively. The calculation of these free energy changes was performed in three distinct stages for both the folded and unfolded systems: (i) decharging of the W3 residue, (ii) van der Waals transformation of W3 into S3 with atomic charges set to zero, and (iii) recharging of the S3 residue. All alchemical transformations proceeded through a series of intermediate hybrid states controlled by the coupling parameter λ, which was varied from 0 to 1 in increments of 0.05. In total, 126 MD simulations were performed to complete the three alchemical transformations for both the folded and unfolded states. System preparation and MD simulations followed a previously established protocol with minor modifications[50]. Specifically, no dummy particles were introduced, and the alchemical transformations for the folded and unfolded states were conducted in separate simulation boxes, as the mutation does not involve a net-charge variation. To improve simulation efficiency, hydrogen mass repartitioning was applied, allowing the use of a 4 fs integration timestep. Cuboid boxes with edges placed at least 9 or 13 Å away from the solute were generated for every system (HCs or capped tripeptides). The HC was simulated in an aqueous environment without the lipid bilayer to reduce computational cost, a simplification justified by the absence of direct interactions between the NTH and membrane lipids. The parameters for the protein and peptide atoms were derived from the ff19SB force field. Table S3 presents a detailed description of the simulation boxes and their composition for each system employed for TI free energy calculations.

For each λ value, 20 ns of productive MD simulation was performed in both folded and unfolded states, with all simulations run in duplicate using distinct initial velocities, assigned during the heating step. To enhance convergence in the folded state during the van der Waals transformation, the simulations at λ values of 0.75, 0.80, and

0.85 were extended to 100 ns, with the λ = 0.80 state further subjected to quadruplicate runs (Fig. S17).

The free energy changes corresponding to the decharging, van der Waals, and recharging steps for both the folded and unfolded states were determined by integrating the respective $\langle \partial V / \partial \lambda \rangle_\lambda$ versus λ profiles obtained from the simulations. Integration was performed using the trapezoidal rule, and the standard errors of the mean (SEM) calculated for each λ point across replicate simulations were propagated to estimate the SEM associated with the resulting free energy values. To ensure equilibration, the initial 25% of frames from each production run were excluded from the analysis.

### Steered molecular dynamics simulations

Steered molecular dynamics (SMD) simulations were performed to introduce one POPC and one CHOL molecule, each positioned within the lipid bilayer, into the open tunnel conformation formed between the TM2–TM3 and TM1–TM2 openings of a wild-type Cx32 monomer in the hemichannel (HC). The simulations were initiated from a trajectory frame exhibiting a fully open TM2–TM3 pathway (Fig. 7E), which had undergone heating and NPT equilibration steps as described previously for conventional MD simulations. To open the tunnel on the TM1–TM2 side, the N-terminal helix (NTH) of this monomer was displaced upward using an angular collective variable (CV) involving residues 3–10, 19–21, and 26–35 (Fig. S18). This CV was gradually increased from 1.09 rad to 1.7 rad over a 50 ns SMD simulation using a harmonic potential with a spring constant of 20 kcal·mol⁻¹·rad⁻², carried out with *pmemd.cuda*.

The final frame of this SMD simulation displayed a fully open tunnel connecting the pore to the bilayer. In separate 50 ns SMD simulations, a CHOL or POPC molecule located near the TM2–TM3 tunnel entrance was pulled into the tunnel. The collective variables were defined as the Euclidean distance between the center carbon atom of each lipid and a dummy particle fixed at the tunnel center. To prevent translational or rotational motions of the entire HC with respect to this dummy particle, a previously described approach requiring the presence of two additional fixed dummy particles was employed[50]. A spring constant of 20 kcal·mol⁻¹·Å⁻² was applied during the pulling process.

### Trajectory analysis

Structural analyses along the trajectories were carried out with different commands of the program *cpptraj* of Amber22[47]. The *secstruct* and *rmsf* commands were employed to monitor the secondary structure and atomic fluctuations of residues within the N-terminal region of Cx32 along the MD trajectories, respectively. Moreover, the *rms and angle* commands were used to calculate root-mean-square deviations for the bound lipids and to monitor the angle formed between the NTH and TM1 during the MD simulations, respectively[47].

Clustering analysis was performed with the *cluster* command of *cpptraj* to determine the representative structure of Cx32 and the bound lipids during the MD simulations. The hierarchical agglomerative method was chosen for clustering and the RMSD the heavy atoms of the bound lipids and of residues belonging to the lipid binding site were used as the metric for clustering[47].

### Reporting summary

Further information on research design is available in the Nature Portfolio Reporting Summary linked to this article.

## Data availability

The MD simulation trajectories and data have been deposited to the OSF repository [https://osf.io/gezpx/]. All coordinates and cryo-EM density maps have been deposited to the Protein Data Bank: 9QN9; 9QND; 9QNF; 9QNT and Electron Microscopy Data Bank: EMD-53240; EMD-53244; EMD-53245; EMD-53250. All other data generated in this study are provided in the Supplementary Information/Source data file. Source data are provided with this paper.

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

## Acknowledgements

J.E.H.G. thanks the National Laboratory for Scientific Computing (LNCC/MCTI, Brazil) for providing HPC resources of the SDumont supercomputer, URL: http://sdumont.lncc.br. We thank the PSI scientific computing team for expert support in high performance computing and image analysis. J.E.H.G. thanks the financial support of the National Council for Scientific and Technological Development (CNPq), Grant 153794/2024-0, and of the Sao Paulo Research Foundation (FAPESP), Grants 2020/08615-8, 2022/03901-8, 2024/13327-2. The work was supported by a Swiss National Science Foundation to V.M.K. (184951).

## Author contributions

Conceptualization: V.M.K. Methodology: P.L., J.E.H.G. and V.M.K. Investigation: P.L., C.F., J.E.H.G. and V.M.K. Visualization: P.L., J.E.H.G. and V.M.K. Funding acquisition: V.M.K. Project administration: V.M.K. Supervision: V.M.K. Writing—original draft: P.L., J.E.H.G. and V.M.K. Writing—review & editing: P.L., J.E.H.G. and V.M.K.

## Competing interests

The authors declare no competing interests.
