## [Transparent Peer Review file · Nature Communications]

Lipid dependence of connexin-32 gap junction channel conformations

Corresponding Author: Professor Volodymyr Korkhov

Version 0:

Reviewer comments:

Reviewer #1

(Remarks to the Author)

Cryo-EM structures of large-pore forming channels have revealed lipidic molecules within the pore, suggesting a novel mechanism in which lipids contribute to channel gating. This manuscript presents high-resolution cryo-electron microscopy (cryo-EM) structures of connexin-32 (Cx32) gap junction channels (GJCs) reconstituted in lipid nanodiscs. The study builds on previous structural work by the authors, which identified lipid densities in Cx32 channels prepared in detergents. The current work identifies three lipid-binding sites (lipids 1–3) and provides evidence suggesting that lipid 2 is likely cholesterol, based on electron density and the structures obtained without cholesterol-hemisuccinate (CHS) during purification. Lipid 3 is located within the pore, potentially obstructing the passage of large molecules. A disease-associated mutation, W3S, appears to exclude both lipid 2 and 3, resulting in a constitutively open channel. These findings suggest a novel lipid-mediated gating mechanism in Cx32.

However, the study reveals major discrepancies between the cryo-EM data and molecular dynamics (MD) simulations. The cryo-EM structures suggest that the W3S mutation excludes lipid 2 while stabilizing the N-terminal helices (NTHs) in an ordered conformation. In contrast, MD simulations predict that NTHs are largely disordered in the W3S mutant and that cholesterol stabilizes NTH. Furthermore, protomer-focused classification of the cryo-EM data shows minimal conformational heterogeneity, whereas MD simulations indicate significant subunit variability. These inconsistencies raise concerns regarding the interpretation and physiological relevance of the lipid densities. The authors acknowledge these limitations but further clarification is warranted.

Major Comments

1. The mechanism by which lipids, particularly lipid 3, exit the pore to allow channel opening is not well explained. Additionally, it is unclear how cholesterol at lipid 2 site exits or redistributes within the membrane environment. If the functional behavior of Cx32 is sensitive to cholesterol levels, could lipid 2 relocate to a non-cholesterol-rich leaflet? Further simulations or experimental data would help support the proposed lipid-gating model.
2. The mechanism by which membrane lipids enter the pore remains speculative. This is a crucial gap, as it directly impacts the plausibility of the lipid-gating hypothesis. Targeted MD simulations could help elucidate whether such lipid infiltration is physically and biologically reasonable.
3. Although previous cryo-EM maps in detergents showed lipid densities in Cx32 hemichannels, this study focuses exclusively on gap junction channels. The absence of hemichannel data from nanodisc preparations makes the interpretation incomplete. Including these structures, or at least discussing them in greater depth, would provide a more holistic understanding of lipid effects across Cx32 assemblies.
4. It is unclear whether lipid 2 is observed in Cx32 structures purified without CHS and reconstituted into LPL nanodiscs. Clarifying this point is essential to assess the relevance of lipid 2 and to rule out CHS-related artifacts.

Minor Comments

1. The manuscript currently relies on qualitative comparisons of MD simulation results. Incorporating statistical analysis would provide a more robust framework for evaluating these results.
2. Line 46: "The potential role of direct of lipid-protein interactions..." should be "The potential role of direct lipid-protein interactions..."
3. Line 99~: The sentence "The NTH transitions from a disordered to an ordered α -helical conformation..." is introduced without sufficient context. A brief description or reference to the disordered state would help the reader understand the significance of this transition.
4. Line 170: "The NTH of Cx32 GJC in POPC..." Please specify that the NTH structure described in POPC refers to the W3S mutant, to avoid confusion.
5. Line 172~: The difference between NTH conformations in the W3S GJC versus the Cx32-W3S hemichannel is noted, but its structural or functional significance is not elaborated. Further explanation would strengthen the discussion.

Reviewer #2

(Remarks to the Author)

The manuscript by Lavriha and colleagues describes a structural biology study of connexin 32 gap junction channel (Cx32GJC) with particular emphasis on the role lipids play in the regulation of this interesting protein. In vivo, Cx32CJG forms oligomeric complex to connect cytoplasm of two neighbouring cells allowing passage of solutes. It is known that lipids play important role in regulating the activity of Cx32CJG and other connexins. In this manuscript authors performed a very careful structural biology study whereby Cx32CJG was reconstituted in lipid nanodiscs. This allowed visualization of another molecule of lipidic origin that, through complex interplay with cholesterol molecule, affected the structural properties of the complex resulting in significantly changed opening of the channel. The structural biology study is supplemented by molecular dynamics characterization of the complex with bound lipids. The study is well written and presented and provides additional insights into the regulation of connexin gap junction proteins by small molecules.

A lot of care was invested to show the presence of the additional lipid (in the manuscript it is referred to as lipid-3) at different experimental conditions. However, I think the main shortcoming of the work is that the density of this molecule is poorly defined in terms of size. Authors assign phospholipid molecule to this density, as it could correspond to the part around glycerol moiety and parts of the acyl chains. The text describing this is carefully used, but I think extra care should be taken not to overinterpret this result at its current stage (i.e. discussion in the introduction section, lines 289-296 on sterols and phospholipids- but these later may not be phospholipids). Could it be that this density could be assigned to some other molecule deriving from the purification procedure and assembly of Cx32CJG into nanodiscs? Could this represent a bent acyl chain? Could the same result be obtained by using their previous method (ref 13) obtaining Cx32CJG oligomers in detergent, but in the presence of some phospholipid(s)?

The interplay between different lipids and structural changes in connexin proteins was previously demonstrated. The results should be, therefore, carefully presented in corresponding parts of the manuscript (abstract, introduction, discussion) and discussed in the view of this previous work, in particular in relation to work from reference 10.

The last paragraph of the discussion describes the effects of gating induced by lipids in other proteins and it is claimed that clear parallels could be drawn by these and their system. But what are these parallels, this is not apparent to me? Is it the same binding to the interior of the channel, affecting the Nterminal region? Please be more specific what these clear parallels are.

Figure 2: what are yellow rectangles in Figure 2A? It should be explained in the legend to figure. If they represent lipid bilayer, are they not too thick (i.e. in comparison to the ones shown in Figure 1B and C)?

Figure 3: What does abbreviation ONS means in panel A, define please.

Figure 3: Panel A shows a comparison between protein in detergent (previous work) and nanodiscs (current work). The PDB Id of the structure used for this panel should be stated in the legend (and the reference).

Figure 7: Panel B of Figure 7 does not contribute anything conceptually and this state of the protein could be described only in the text.

Materials and methods section: what was the source of POPC and LPL (lines 19,20)? Please define.

The purification of connexin (lines 51-66) is not clear to me. Why do you need anti-GFP nanobody coupled CNBr Sepharose for purification? Does the construct contains GFP with appropriate cleavage site (as in the next step something is cleaved off)? Please be more specific here about the construct that was used for the study.

Figure S1: There are two bands for Cx32 on gels shown in Figure S1 in A,B and no protein in panel C.. Please comment.

Legend to Figure S8 does not correspond to the figure.

Reviewer #3

(Remarks to the Author)

The manuscript "Lipid dependence of connexin-32 gap junction channel conformations" by Lavriha et al. describes a series of cryoEM structures of Connexin-32 wildtype and the disease-associated mutation W3S. The structures provide insights into the gating mechanism of connexin channels, supporting a role of sterols and phospholipids to directly bind to the inner channel pore, thereby stabilizing the N-terminal helix in a closed channel conformation. Accompanying molecular dynamics simulations strengthen the idea that phospholipids and sterols dock to different sites in the inner pore of Connexin-32 and stabilize the N-terminal helix. The manuscript is overall very well written and easy to follow. The authors present a thorough study, which focuses on a topic of wide interest in this area, and which highlights the importance of lipids in regulating gap junction channels. I only have minor points that should be addressed by the authors:

1. The authors describe that the Cx32 wildtype structure was used as the starting structure for all MD simulations. However, the authors should also include how they introduced the mutation into the system prior to the MD simulations, and how they decided, which rotamer of serine to choose, so that the observed destabilization of the N-terminal helix is not a result of the choice of serine rotamer in that local area.

2. A key effect of the binding of sterols and phospholipids to the channel pore appears to be the support of the N-terminal helix to rearrange from the open to the NO state. Such conformational changes occur over long timescales, and are not captured by the performed 1 μ s simulations. Other studies have shown that e.g. Hamiltonian Replica Exchange MD are required to compute the free energy of the open-to-closed transition (e.g. Mhashal et al., ACS Catalysis, 2020).

Version 1:

Reviewer comments:

Reviewer #1

(Remarks to the Author)

The authors have adequately addressed the concerns raised by the reviewers. The revised manuscript includes additional principal component analysis and provides clearer explanations for previously ambiguous points, enhancing the transparency and comprehensiveness of the study. This is an interesting and well-executed study, and I have no further comments.

Reviewer #2

(Remarks to the Author)

I have no further comments, the authors have addressed all my comments.

Reviewer #3

(Remarks to the Author)

The authors have addressed all my comments raised in the previous revision. The revised paper has improved and I have no further concerns.

RESPONSE TO REVIEWER'S COMMENTS

Reviewer #1 (Remarks to the Author):

Cryo-EM structures of large-pore forming channels have revealed lipidic molecules within the pore, suggesting a novel mechanism in which lipids contribute to channel gating. This manuscript presents high-resolution cryo-electron microscopy (cryo-EM) structures of connexin-32 (Cx32) gap junction channels (GJCs) reconstituted in lipid nanodiscs. The study builds on previous structural work by the authors, which identified lipid densities in Cx32 channels prepared in detergents. The current work identifies three lipid-binding sites (lipids 1–3) and provides evidence suggesting that lipid 2 is likely cholesterol, based on electron density and the structures obtained without cholesterol-hemisuccinate (CHS) during purification. Lipid 3 is located within the pore, potentially obstructing the passage of large molecules. A disease-associated mutation, W3S, appears to exclude both lipid 2 and 3, resulting in a constitutively open channel. These findings suggest a novel lipid-mediated gating mechanism in Cx32.

However, the study reveals major discrepancies between the cryo-EM data and molecular dynamics (MD) simulations. The cryo-EM structures suggest that the W3S mutation excludes lipid 2 while stabilizing the N-terminal helices (NTHs) in an ordered conformation. In contrast, MD simulations predict that NTHs are largely disordered in the W3S mutant and that cholesterol stabilizes NTH. Furthermore, protomer-focused classification of the cryo-EM data shows minimal conformational heterogeneity, whereas MD simulations indicate significant subunit variability. These inconsistencies raise concerns regarding the interpretation and physiological relevance of the lipid densities. The authors acknowledge these limitations but further clarification is warranted.

Major Comments

1. The mechanism by which lipids, particularly lipid 3, exit the pore to allow channel opening is not well explained. Additionally, it is unclear how cholesterol at lipid 2 site exits or redistributes within the membrane environment. If the functional behavior of Cx32 is sensitive to cholesterol levels, could lipid 2 relocate to a non-cholesterol-rich leaflet? Further simulations or experimental data would help support the proposed lipid-gating model.

2. The mechanism by which membrane lipids enter the pore remains speculative. This is a crucial gap, as it directly impacts the plausibility of the lipid-gating hypothesis. Targeted MD simulations could help elucidate whether such lipid infiltration is physically and biologically reasonable.

RESPONSE: We are grateful to the reviewer for these questions. The points 1 and 2 are related, so we consolidated our answers to both questions here. Following these comments of the reviewer we have initiated MD-based analysis of the putative lateral access movement of the lipid molecules into the pore. We provide here a possible explanation of how the lipids may be able to access their cognate binding sites. We should stress that these MD simulations are at the moment not directly supported by experiments, as this presents a distinct challenge in itself. However, the observations now included in the revised manuscript (please see below) are an important first step towards a better understanding of the lateral lipid movement in connexin channels.

To elucidate potential pathways for lipid entrance, we conducted principal component analysis (PCA) to capture the most significant motions of the C α atoms of the Cx32 monomer, using data from all simulations of wild-type Cx32 HCs presented in this study (see Figures 7A-C in the revised manuscript). In total, the analysis encompassed 6 chains per system \times 2 replicate simulations \times 4 systems containing Cx32 wild-type \times 1 μ s per simulation, amounting to 48 μ s of simulation time. As described in the updated manuscript, the dominant motions associated with the first two principal components correspond to: (1)

vertical displacement of the N-terminal helix (NTH) relative to the membrane plane, and (2) the lateral displacement of TM2 relative to TM3. These two movements could facilitate the formation of a transient lateral access pathway through which cholesterol and POPC might, in principle, enter (see Figures 7D-E in the resubmitted manuscript). Pulling simulations indicate that lipid entrance would require a greater vertical displacement of the NTH than what was observed in conventional MD simulations of the Cx32 wt HCs (Figure 7F). Nevertheless, it can be hypothesized that the initial insertion of a lipid between TM2 and TM3 may trigger a wider vertical movement of the NTH. It is important to note, however, that this hypothesis needs to be experimentally validated. Moreover, our simulations are limited by the absence of the flexible TM2–TM3 linker, which is unresolved in available structural data and was therefore not modeled. The presence of this linker could potentially constrain the separation between TM2 and TM3 observed in our simulations. These aspects are detailed in the revised version of the manuscript.

3. Although previous cryo-EM maps in detergents showed lipid densities in Cx32 hemichannels, this study focuses exclusively on gap junction channels. The absence of hemichannel data from nanodisc preparations makes the interpretation incomplete. Including these structures, or at least discussing them in greater depth, would provide a more holistic understanding of lipid effects across Cx32 assemblies.

RESPONSE: We are grateful to the reviewer for highlighting this discrepancy. This prompted us to improve the description of the outcome of nanodisc reconstitutions. We were essentially unable to obtain the hemichannel reconstructions for the lipid-reconstituted Cx32 channels. While this was possible for the detergent samples, upon nanodisc reconstitution the remaining particles that could lead to a 3D reconstruction were predominantly those of gap junction channels. We clarified this in the revised manuscript (page 4, line 108):

“It is noteworthy that, unlike the Cx32 sample that previously led to cryo-EM reconstructions of both GJCs and HCs in detergent micelles¹³, upon nanodisc reconstitution we could observe predominantly the full GJCs. This may reflect a greater stability of the fully assembled GJCs, allowing these complexes to stay intact during detergent removal. In contrast, HCs may suffer in a number of ways during the lipid reconstitution procedure, which results in very few particles that can be used for cryo-EM image processing and structure determination of HCs in nanodiscs. Therefore our study was focused on the Cx32 GJCs.”

4. It is unclear whether lipid 2 is observed in Cx32 structures purified without CHS and reconstituted into LPL nanodiscs. Clarifying this point is essential to assess the relevance of lipid 2 and to rule out CHS-related artifacts.

RESPONSE: As we observed virtually identical results with the POPC- and LPL-reconstituted Cx32 samples, we opted to focus on the POPC-reconstituted nanodiscs only when it came to the samples where CHS was omitted during the purification: LPL reconstitutions were not performed in the absence of CHS. This was in part motivated by the very poor basal yields of purified protein upon omission of CHS from the purification procedure, which is not uncommon for membrane proteins. Performing the experiments in the presence of a chemically defined lipid species, POPC, appeared to be the most straightforward approach producing interpretable results.

Minor Comments

1. The manuscript currently relies on qualitative comparisons of MD simulation results. Incorporating statistical analysis would provide a more robust framework for evaluating these results.
RESPONSE: We have now included the statistical analysis of the key parameters resulting from the MD simulation analysis (Table S1). The probability of adopting an α helix or an alternative conformation (labeled as Other in the new figures 5 and S14) was assessed for every NTH in the HCs throughout the two replicate simulations conducted for every system. Averaging of these probabilities over the course of the simulations (discarding the first 600 ns) produced 12 probability values for the segment occurring as an alpha helix and 12 values for that occurring as a random coil. We assessed the

normality of the data and then applied appropriate statistical tests to compare the probabilities in a pairwise fashion for the systems of interest in different conditions (Table S1). As shown in the updated version of the manuscript, the probability differences between Cx32 wt and Cx32 W3S are statistically significant ($p < 0.05$).

We would like to stress that the secondary structure probabilities shown in the graphs correspond specifically to residues 3-5 and 9-11, at both ends of the NTH, rather than to the full 3-10, as wrongly indicated in the previous version. Residues 6, 7, and 8 were deliberately excluded from the analysis because the center of the alpha helix exhibited less conformational variability, as observed especially in Figure S15A. Their inclusion would have biased the average values toward higher helicity and thus masked the destabilizing effect of the mutation on both NTH ends. The figure captions have been revised to reflect these clarifications. Moreover, differently from the previous versions of Figures 5 and S14, the updated figures use a binary classification of the secondary structure as (1) α helix or (2) other, which groups all other categories. By doing so, the bar heights are the same in all of the analyzed time intervals, which makes it easier to detect differences in the probability of the selected residues adopting an α -helical conformation in the studied systems.

In addition, to further evaluate the impact of the W3S mutation on NTH stability, we performed thermodynamic integration free energy calculations. These calculations were carried out for a thermodynamic cycle involving the W3S mutation both in the context of the hemichannel NTH and in a capped peptide (ACE-Gly-X-Gly-NME, where X is either Trp or Ser). This linear peptide models the unfolded state. The resulting $\Delta\Delta G$ of approximately -3.62 kcal/mol supports a clear stabilization of the folded (α -helical) state over the unfolded state by Trp relative to Ser. The new results and the methodology are included in the revised manuscript (Table S2 and Figure S17).

2. Line 46: "The potential role of direct of lipid-protein interactions..." should be "The potential role of direct lipid-protein interactions..."

RESPONSE: We have corrected this line accordingly (page 2, line 46).

3. Line 99~: The sentence "The NTH transitions from a disordered to an ordered α -helical conformation..." is introduced without sufficient context. A brief description or reference to the disordered state would help the reader understand the significance of this transition.

RESPONSE: We thank the reviewer for spotting this ambiguous statement. This may be a semantic distinction, but indeed we can not claim that NTH is "disordered" – it may be better to state "flexible" or "unresolved". It is indeed possible that NTH remains alpha-helical while being flexible. To more accurately reflect our observations, this sentence now reads (page 3, line 99):

"The NTH transitions from a flexible structural element unresolved in our cryo-EM reconstruction of Cx32 GJC in detergent ¹³ to a well ordered..".

4. Line 170: "The NTH of Cx32 GJC in POPC..." Please specify that the NTH structure described in POPC refers to the W3S mutant, to avoid confusion.

RESPONSE: We have updated the text accordingly (page 6, line 181):

"The NTH of W3S GJC in POPC is ordered and points towards the inside of the channel pore, in a conformation slightly different from that in the wild-type Cx32 maintaining a pore diameter of >10 Å consistent with an open channel (Figure 4D-F, S12)."

5. Line 172~: The difference between NTH conformations in the W3S GJC versus the Cx32-W3S hemichannel is noted, but its structural or functional significance is not elaborated. Further explanation would strengthen the discussion.

RESPONSE: We have added the following text into the corresponding part of the discussion (page 6, line 185):

“It is important to note that while structurally the NTH conformations of the W3S GJC in POPC nanodiscs and W3S HC in detergent are distinct, the two states of the mutated protein are also dramatically distinct in their functional properties. Whereas the GJC function of the W3S mutant appears to be very similar to that of the wild-type Cx32, the HC activity of the W3S is dramatically reduced¹³. The current limitations of our protein preparations prohibit HC structure determination, and we are at loss to explain the reason why Cx32HCs are refractory to structure determination upon lipid reconstitution. The optimal approach to correlate the structural and functional properties of W3S GJCs and HCs would be to compare the corresponding structures determined for the proteins reconstituted in the same lipidic environment. Stabilization of wild-type and W3S mutant Cx32 HCs in nanodiscs for cryo-EM structure determination will require careful future experimentation.”

Reviewer #2 (Remarks to the Author):

The manuscript by Lavriha and colleagues describes a structural biology study of connexin 32 gap junction channel (Cx32GJC) with particular emphasis on the role lipids play in the regulation of this interesting protein. In vivo, Cx32CJG forms oligomeric complex to connect cytoplasm of two neighbouring cells allowing passage of solutes. It is known that lipids play important role in regulating the activity of Cx32CJG and other connexins. In this manuscript authors performed a very careful structural biology study whereby Cx32CJG was reconstituted in lipid nanodiscs. This allowed visualization of another molecule of lipidic origin that, through complex interplay with cholesterol molecule, affected the structural properties of the complex resulting in significantly changed opening of the channel. The structural biology study is supplemented by molecular dynamics characterization of the complex with bound lipids. The study is well written and presented and provides additional insights into the regulation of connexin gap junction proteins by small molecules.

A lot of care was invested to show the presence of the additional lipid (in the manuscript it is referred to as lipid-3) at different experimental conditions. However, I think the main shortcoming of the work is that the density of this molecule is poorly defined in terms of size. Authors assign phospholipid molecule to this density, as it could correspond to the part around glycerol moiety and parts of the acyl chains. The text describing this is carefully used, but I think extra care should be taken not to overinterpret this result at its current stage (i.e. discussion in the introduction section, lines 289-296 on sterols and phospholipids- but these later may not be phospholipids). Could it be that this density could be assigned to some other molecule deriving from the purification procedure and assembly of Cx32CJG into nanodiscs? Could this represent a bent acyl chain? Could the same result be obtained by using their previous method (ref 13) obtaining Cx32CJG oligomers in detergent, but in the presence of some phospholipid(s)?

RESPONSE: This is a great point by the reviewer. We thought along the same lines as the reviewer, pondering the different possibilities. In the end we decided that the most likely explanation for what we have seen in our structures were six partially ordered phospholipid molecules that roughly match the bifurcated densities in the vicinity of the NTH regions. This seemed to be the most obvious explanation at the time – and it still is. However, the reviewer is right – it is entirely possible that this density feature could potentially match a bent acyl chain, or yet another molecule that co-elutes along with the protein. The simple logic of interpreting this density as the one lipid species that we have supplemented during reconstitution makes sense, however we agree that this needs to be worded even more carefully, as in the revised text (page 5, line 152):

“While our interpretation of the observed densities is that they correspond to the bound phospholipid molecules, we can not completely exclude a possibility that these densities may also correspond to bent acyl chain of the lipids, or even some other small molecule species present in the sample (co-eluting with the purified protein and appearing as ordered density upon addition of the lipids and removal of the detergent).”

Regarding the idea of mixing the lipid into the detergent sample, while it does seem like a potentially attractive approach, we opted to stay clear of it. The whole point of our study was to exclude the mixed population of lipids and detergents, and we reasoned at the design stage that we risk obscuring the

interpretation even more by mixing lipids in without removing the detergents completely from the sample.

The interplay between different lipids and structural changes in connexin proteins was previously demonstrated. The results should be, therefore, carefully presented in corresponding parts of the manuscript (abstract, introduction, discussion) and discussed in the view of this previous work, in particular in relation to work from reference 10.

RESPONSE: We are grateful to the reviewer for pointing this out. The excellent study described reference 10 was on Cx43. This is a different homologue of the connexin family, with potentially different regulatory mechanisms. Following the suggestion of the reviewer we have added new text in the discussion, to reflect the previous work. The added text reads as follows (page 11, line 419):

“For example, a comprehensive structural analysis of the conformational changes in Cx43 revealed the existence of distinct conformation states of the NTH: the gate-covering, pore lining, and flexible intermediate NTH states¹⁰. Moreover, Lee et al. found that the presence of CHS, varied pH and use of a C-terminally truncated Cx43 construct can influence the NTH conformation¹⁰. It remains to be determined ..”

The last paragraph of the discussion describes the effects of gating induced by lipids in other proteins and it is claimed that clear parallels could be drawn by these and their system. But what are these parallels, this is not apparent to me? Is it the same binding to the interior of the channel, affecting the Nterminal region? Please be more specific what these clear parallels are.

RESPONSE: This is a good point, and there is a certain risk in going deep in this argument. Any detailed comparisons at this stage would be too speculative. The updated manuscript now features a minimally speculative final statement (page 12, line 392):

“Lipid-mediated regulation has emerged as a common denominator in those studies. We anticipate that future investigations probing the molecular mechanisms of the large pore ion channel family members will reveal the common mechanistic features of the lipid-mediated modulation of these proteins’ structure and function, which may reflect our own observations on lipid-reconstituted Cx32 channels.”

Figure 2: what are yellow rectangles in Figure 2A? It should be explained in the legend to figure. If they represent lipid bilayer, are they not too thick (i.e. in comparison to the ones shown in Figure 1B and C)?

RESPONSE: The pale yellow rectangles indeed roughly correspond to the two lipid bilayers. We adjusted this slightly, to make the rectangle boundaries more symmetrical. These rectangles are largely meant a visual aid to help the reader orient themselves with respect to the view of the protein displayed in the figure. We hope that in the current adjusted form this visual aid is acceptable.

Figure 3: What does abbreviation ONS means in panel A, define please.

RESPONSE: We have updated the figure – this should be “NO-state” (“ONS” was an older working abbreviation in this figure that we overlooked). We have adjusted the figure accordingly.

Figure 3: Panel A shows a comparison between protein in detergent (previous work) and nanodiscs (current work). The PDB Id of the structure used for this panel should be stated in the legend (and the reference).

RESPONSE: We have updated the Figure 3 legend accordingly:

“Pore representations of Cx32 GJC solved in detergent (PDB ID: 7ZXM¹³) ..”

Figure 7: Panel B of Figure 7 does not contribute anything conceptually and this state of the protein could be described only in the text.

RESPONSE: We understand the reviewer's argument, and we could potentially exclude this panel. We opted now to keep it, for the reasons outlined below (we hope the reviewer will agree with our choice). This figure was not meant both as the mechanistic insight into the lipid-lipid or lipid-protein interactions of Cx32, as well as a visual recapitulation of our experimental observations. In principle we could exclude this panel from the manuscript without a major loss to the overall message. However, we found that this figure works very well to explain and summarise our story. For example, we found the complete figure, including panels A and B to be very instrumental in explaining the results during a recent Gordon Research Conference on lipid biology (Waterville Valley, July 2025), where these results were presented as a poster (VMK). Based on this, we think including this panel may be very helpful for a general reader to get an overview of the results in a simple and accessible format.

Materials and methods section: what was the source of POPC and LPL (lines 19,20)? Please define.
RESPONSE: The lipids were purchased from Avanti Polar Lipids, and we have updated the methods with this information (Page 2, line 20).

The purification of connexin (lines 51-66) is not clear to me. Why do you need anti-GFP nanobody coupled CNBr Sepharose for purification? Does the construct contains GFP with appropriate cleavage site (as in the next step something is cleaved off)? Please be more specific here about the construct that was used for the study.

RESPONSE: We have clarified this in the methods. Indeed anti-GFP nanobody is a powerful tool we use for purification of almost all membrane proteins in our lab – this has in many cases produced outstanding results, as the nanobody is highly specific and often superior to other established affinity purification reagents. The method relies on capturing the fluorescent protein-tagged protein on the nanobody resin, followed by elution by 3C protease cleavage (a 3C site is located between Cx32 and the C-terminal YFP in our construct). Page 3, line 52:

“The adherent HEK293F cells were transfected with a pACMV plasmid encoding Cx32 or Cx32-W3S mutant, in frame with a C-terminal 3C-YFP-twinStrep tag. Following a 48 h period after transfection, cells harvested from 100 15-cm culture plates were resuspended in buffer A ..”

Figure S1: There are two bands for Cx32 on gels shown in Figure S1 in A,B and no protein in panel C..
Please comment.

RESPONSE: The presence of more than one bands Cx32 is not unusual, as we do not heat our membrane proteins and thus some of the protein may retain partial structure in the presence of SDS micelles during SDS PAGE. Non-uniform migration of membrane proteins on SDS PAGE is not uncommon, as they do not behave the same as globular soluble proteins (for which one would typically heat the samples before loading on the SDS PAGE gel).

The panel C does contain Cx32 protein, but it appears that the sample also contains an excess of MSP2N2, which manifests in a much stronger staining of the MSP2N2 band by coomassie blue in comparison. This exact sample was used for structure determination, and thus we have included this gel image. There can be two possible explanations: (1) An excess of MSP2N2 was introduced in this case during reconstitution due to the operator error; (2) Identical MSP2N2 ratios were used for this protein as for the wild-type Cx32 proteins in panels A, B and D, but W3S protein may have been much less stable under the reconstitution conditions, resulting in some of the protein precipitating, attaching to the Biobeads, etc. Either of the two options may apply here.

Legend to Figure S8 does not correspond to the figure.

RESPONSE: We are very grateful to the reviewer for spotting this, and we have corrected this in the revised version. The new legend reads as follows (page 17, line 299):

“**Figure S8. Effect of lipid-2 on Cx32 GJC NTH.** (A) Lipid-2 density can accommodate cholesterol, or cholesteryl hemisuccinate (CHS), and digitonin, used during protein purification. (B) Cx32 GJC reconstituted in LPL-containing nanodisc has the similar densities as Cx32 GJC, reconstituted in POPC

containing nanodisc. (C) Densities of lipid-2 and lipid-3 in the cryo-EM map of the Cx32 GJC in LPL-containing nanodiscs. (D) HOLE analysis of the pore conduction pathway of Cx32 GJC in LPL-containing nanodiscs. The pathway colored in yellow includes POPC in the calculation, whereas the grey excludes it. The arrows represent the points of pore constriction due to NTH rearrangement.“

Reviewer #3 (Remarks to the Author):

The manuscript “Lipid dependence of connexin-32 gap junction channel conformations” by Lavriha et al. describes a series of cryoEM structures of Connexin-32 wildtype and the disease-associated mutation W3S. The structures provide insights into the gating mechanism of connexin channels, supporting a role of sterols and phospholipids to directly bind to the inner channel pore, thereby stabilizing the N-terminal helix in a closed channel conformation. Accompanying molecular dynamics simulations strengthen the idea that phospholipids and sterols dock to different sites in the inner pore of Connexin-32 and stabilize the N-terminal helix. The manuscript is overall very well written and easy to follow. The authors present a thorough study, which focuses on a topic of wide interest in this area, and which highlights the importance of lipids in regulating gap junction channels. I only have minor points that should be addressed by the authors:

1. The authors describe that the Cx32 wildtype structure was used as the starting structure for all MD simulations. However, the authors should also include how they introduced the mutation into the system prior to the MD simulations, and how they decided, which rotamer of serine to choose, so that the observed destabilization of the N-terminal helix is not a result of the choice of serine rotamer in that local area.

RESPONSE: We thank the reviewer for spotting this, and we have updated the methods section accordingly.

The section: “System setup for molecular dynamics simulations” in the resubmitted version of SI includes the following sentence at the end: “It is worth noting that the initial structures to generate the topology and coordinate files of systems involving the W3S mutation were derived from the equilibrated structures (see below) of the wild-type systems, by manually renaming the W3 residue to SER in the PDB files and deleting the side-chain atoms beyond the CB. The missing hydroxyl group of the Ser residue was added automatically by *tleap*”.

Moreover, we have included the following information in the main text: “This underscores the instability of the NTHs carrying the W3S mutation, which were initially modeled by retaining the C β orientation of W3 in the wild-type protein” (page 7, line 220).

By following this approach, we guaranteed that the deviations from the α -helical conformation observed over the course of the MD simulations stemmed from the nature of the new residue (Ser) and not as a consequence of having chosen a rotameric form that destabilizes the NTH.

2. A key effect of the binding of sterols and phospholipids to the channel pore appears to be the support of the N-terminal helix to rearrange from the open to the NO state. Such conformational changes occur over long timescales, and are not captured by the performed 1 μ s simulations. Other studies have shown that e.g. Hamiltonian Replica Exchange MD are required to compute the free energy of the open-to-closed transition (e.g. Mhashal et al., ACS Catalysis, 2020).

RESPONSE: This is an intriguing point, which we considered carefully. Our new PCA results show that the NTH vertical motion that can lead to open and NO states are captured during the microsecond-long MD simulations (Figure 7C). Therefore, there is in principle no need to use enhanced sampling techniques for that purpose in this particular system. Moreover, the angle histograms observed in Fig. S18 to S22 (Figures S17 to 21 in the previous SI version) already suggested the occurrence of significant vertical motions of the NTH relative to TM1. However, the inclusion of PCA in the new version, further reinforces this point. Based on this, in the current revised version we have opted to omit the proposed reference, although we could probably find a way to integrate the reference to Mhashal et al., 2020, in case the reviewer deems this essential.